# CAMERAS AS RAYS:
# POSE ESTIMATION VIA RAY DIFFUSION

**Jason Y. Zhang**,\***Amy Lin**\*, **Moneish Kumar,**
**Tzu-Hsuan Yang, Deva Ramanan, Shubham Tulsiani**
Carnegie Mellon University

## ABSTRACT

Estimating camera poses is a fundamental task for 3D reconstruction and remains challenging given sparsely sampled views ($<10$). In contrast to existing approaches that pursue top-down prediction of global parametrizations of camera extrinsics, we propose a distributed representation of camera pose that treats a camera as a bundle of rays. This representation allows for a tight coupling with spatial image features improving pose precision. We observe that this representation is naturally suited for set-level transformers and develop a regression-based approach that maps image patches to corresponding rays. To capture the inherent uncertainties in sparse-view pose inference, we adapt this approach to learn a denoising diffusion model which allows us to sample plausible modes while improving performance. Our proposed methods, both regression- and diffusion-based, demonstrate state-of-the-art performance on camera pose estimation on CO3D while generalizing to unseen object categories and in-the-wild captures.

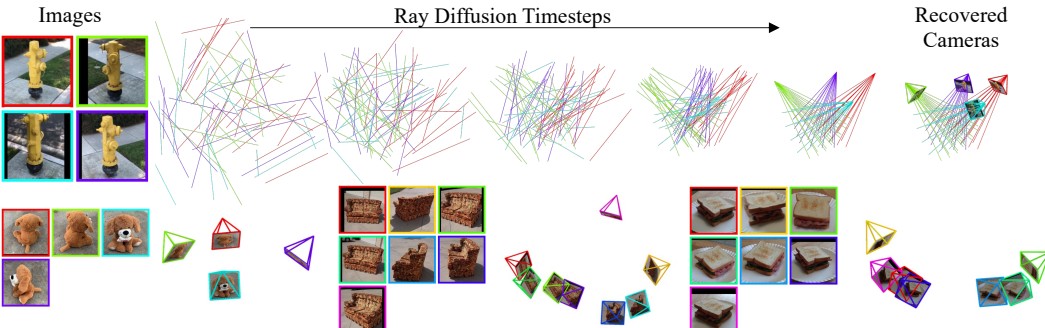

Figure 1: **Recovering Sparse-view Camera Parameters by Denoising Rays.** *Top:* Given sparsely sampled images, our approach learns to denoise camera rays (represented using Plücker coordinates). We then recover camera intrinsics and extrinsics from the positions of the rays. *Bottom:* We demonstrate the generalization of our approach for both seen (teddybear) and unseen object categories (couch, sandwich).

## 1 INTRODUCTION

We have witnessed rapid recent progress toward the goal of obtaining high-fidelity 3D representations given only a sparse set of input images (Zhang et al., 2021; Goel et al., 2022; Zhou & Tulsiani, 2023; Long et al., 2022; Cerkezi & Favaro, 2024; Truong et al., 2023). However, a crucial requirement is the availability of camera poses corresponding to the 2D input images. This is particularly challenging as structure-from-motion methods fail to reliably infer camera poses under settings with sparsely sampled views (also referred to as sparse-view or wide-baseline in prior works). To fill this performance gap, recent learning-based approaches have examined the task of predicting cameras given a sparse set of input images, and investigated regression (Jiang et al., 2024; Sinha et al., 2023), energy-based modeling (Zhang et al., 2022; Lin et al., 2024) and denoising diffusion (Wang et al.,

---

\*denotes equal contribution. Project Page: https://jasonyzhang.com/RayDiffusion.

2023) for inference. However, while exploring a plethora of learning techniques, these methods have largely side-stepped a crucial question: *what representation of camera poses should learning-based methods predict*?

At first, there may seem to be an obvious answer. After all, every student of projective geometry is taught that (extrinsic) camera matrices are parameterized with a single rotation and a translation. Indeed, all of the above-mentioned methods adapt this representation (albeit with varying rotation parametrizations *e.g.,* matrices, quaternions, or angles) for predicting camera poses. However, we argue that such a parsimonious *global* pose representation maybe suboptimal for neural learning, which often benefits from over-parameterized *distributed* representations. From a geometric perspective, classical bottom-up methods benefit from low-level correspondence across pixels/patches, while learning-based methods that predict global camera representations may not easily benefit from such (implicit or explicit) associations.

In this work, we propose an alternate camera parametrization that recasts the task of pose inference as that of patch-wise ray prediction (Fig. 1). Instead of predicting a global rotation and global translation for each input image, our model predicts a separate ray passing through each patch in each input image. We show that this representation is naturally suited for transformer-based set-to-set inference models that process sets of features extracted from image patches. To recover the camera extrinsics ($R$, $t$) and intrinsics ($K$) corresponding to a classical perspective camera, we optimize a least-square objective given the predicted ray bundle. It is worth noting that the predicted ray bundle itself can be seen as an encoding of a *generic camera* as introduced in Grossberg & Nayar (2001), which can capture non-perspective cameras such as catadioptric imagers or orthographic cameras whose rays may not even intersect at a center of projection.

We first illustrate the effectiveness of our distributed ray representation by training a patch-based transformer with a standard regression loss. We show that this already surpasses the performance of state-of-the-art pose prediction methods that tend to be much more compute-heavy (Lin et al., 2024; Sinha et al., 2023; Wang et al., 2023). However, there are natural ambiguities in the predicted rays due to symmetries and partial observations (Zhang et al., 2022; Wang et al., 2023). We extend our regression-based method to a denoising diffusion-based probabilistic model and find that this further improves the performance and can recover distinct distribution modes. We demonstrate our approach on the CO3D dataset (Reizenstein et al., 2021) where we systematically study performance across seen categories as well as generalization to unseen ones. Moreover, we also show that our approach can generalize even to unseen datasets and present qualitative results on in-the-wild self-captures. In summary, our contributions are as follows:

- We recast the task of pose prediction as that of inferring per-patch ray equations as an alternative to the predominant approach of inferring global camera parametrizations.
- We present a simple regression-based approach for inferring this representation given sparsely sampled views and show even this simple approach surpasses the state-of-the-art.
- We extend this approach to capture the distribution over cameras by learning a denoising diffusion model over our ray-based camera parametrization, leading to further performance gains.

## 2 RELATED WORK

### 2.1 STRUCTURE-FROM-MOTION AND SLAM

Both Structure-from-motion and SLAM aim to recover camera poses and scene geometry from a large set of unordered or ordered images. Classic SfM (Snavely et al., 2006) and indirect SLAM (Mur-Artal et al., 2015; Mur-Artal & Tardós, 2017; Campos et al., 2021) methods generally rely on finding correspondences (Lucas & Kanade, 1981) between feature points (Bay et al., 2006; Lowe, 2004) in overlapping images, which are then efficiently optimized (Schönberger & Frahm, 2016; Schönberger et al., 2016) into coherent poses using Bundle Adjustment (Triggs et al., 1999). Subsequent work has improved the quality of features (DeTone et al., 2018), correspondences (Shen et al., 2020; Yang & Ramanan, 2019; Sarlin et al., 2020), and the bundle adjustment process itself (Tang & Tan, 2019; Lindenberger et al., 2021). On the contrary, rather than minimize geometric reconstruction errors, direct SLAM methods (Davison et al., 2007; Schops et al., 2019)

Figure 2: **Converting Between Camera and Ray Representations.** We represent cameras as a collection of 6-D Plücker rays consisting of directions and moments. We convert the traditional representation of cameras to the ray bundle representation by unprojecting rays from the camera center to pixel coordinates. We convert rays back to the traditional camera representation by solving least-squares optimizations for the camera center, intrinsics matrix, and rotation matrix. See Sec. 3.1 for more details.

optimize photometric errors. While the methods described in this section can achieve (sub)pixel-perfect accuracy, their reliance on dense images is unsuitable for sparse-view pose estimation.

## 2.2 POSE ESTIMATION FROM SPARSELY SAMPLED VIEWS

Estimating poses from sparsely sampled images (also called sparse-view or wide-baseline pose estimation in prior work) is challenging as methods cannot rely on sufficient (or even any) overlap between nearby images to rely on correspondences. The most extreme case of estimating sparse-view poses is recovering the relative pose given 2 images. Recent works have explored how to effectively regress relative poses (Balntas et al., 2018; Rockwell et al., 2022; Cai et al., 2021) from wide-baseline views. Other works have explored probabilistic approaches to model uncertainty when predicting relative pose (Zhang et al., 2022; Chen et al., 2021).

Most related to our approach are methods that can predict poses given multiple images. Rel-Pose (Zhang et al., 2022) and RelPose++ (Lin et al., 2024) use energy-based models to compose relative rotations into sets of camera poses. SparsePose (Sinha et al., 2023) learns to iteratively refine sparse camera poses given an initial estimate, while FORGE (Jiang et al., 2024) exploits synthetic data to learn camera poses. The most comparable to us is PoseDiffusion (Wang et al., 2023), which also uses a diffusion model to denoise camera poses. However, PoseDiffusion denoises the camera parameters directly, whereas we denoise camera rays which we demonstrate to be more precise. Concurrently to our work, PF-LRM (Wang et al., 2024a) and DUSt3R (Wang et al., 2024b) predict sparse poses by predicting pixel-aligned pointclouds (as opposed to rays in our work) and using PnP to recover cameras.

## 2.3 RAY-BASED CAMERA PARAMETERIZATIONS

Prior work in calibrating generic camera representations has used ray-based representations of cameras, mainly for fish-eyed lenses for which the pinhole model is not a good approximation (Kannala & Brandt, 2006). Grossberg & Nayar (2001); Dunne et al. (2010) consider the most general camera model, where each pixel projection is modeled by its ray. Even with better algorithms (Schops et al., 2020), the large number of parameters in these camera models makes calibration difficult. Although these works also make use of ray-based camera representations, their focus is on calibration (intrinsics) and require known calibration patterns. Neural Ray Surfaces (Vasiljevic et al., 2020) considers learning the poses of generic cameras but does so from video rather than sparse views.

Parameterizing viewpoints using camera rays is also commonly used in the novel view synthesis community. Rather than render a full image at once, the pixel-wise appearance is conditioned per ray (Mildenhall et al., 2020; Sitzmann et al., 2021; Watson et al., 2023) given known cameras. In contrast, we aim to recover the camera itself.

## 3 METHOD

Our aim is to recover cameras from a sparse set of images $\{I_1, \ldots, I_N\}$. Rather than predict global camera parametrizations directly as done in previous work, we propose a ray-based representation

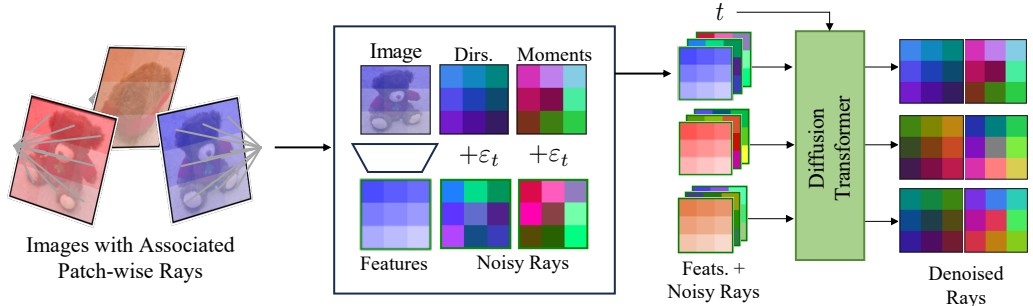

Figure 3: **Denoising Ray Diffuser Network**. Given a noisy ray corresponding to an image patch, our denoising ray diffusion model predicts the denoised ray. We concatenate spatial image features (Oquab et al., 2023) with noisy rays, represented with 6-dimensional Plücker coordinates (Plücker, 1828) that are visualized as 3-channel direction maps and 3-channel moment maps. We use a transformer to jointly process all image patches and associated noisy rays to predict the original denoised rays.

that can be seamlessly converted to and from the classic representation (Sec. 3.1). We then describe a regression-based architecture to predict ray-based cameras in Sec. 3.2. We build on this architecture to introduce a probabilistic framework that estimates the rays using diffusion to handle uncertainties and symmetries that arise from sparsely sampled views in Sec. 3.3.

## 3.1 REPRESENTING CAMERAS WITH RAYS

**Distributed Ray Representation.** Typically, a camera is parameterized by its extrinsics (rotation $\mathbf{R} \in \mathrm{SO}(3)$, translation $\mathbf{t} \in \mathbb{R}^3$) and intrinsics matrix $\mathbf{K} \in \mathbb{R}^{3 \times 3}$. Although this parameterization compactly relates the relationship of world coordinates to pixel coordinates using camera projection ($\mathbf{u} = \mathbf{K}[\mathbf{R} \mid \mathbf{T}]\mathbf{x}$), we hypothesize that it may be difficult for a neural network to directly regress this low-dimensional representation. Instead, inspired by generalized camera models (Grossberg & Nayar, 2001; Schops et al., 2020) used for calibration, we propose to *over-parameterize* a camera as a collection of rays:

$$\mathcal{R} = \{\mathbf{r}_1, \ldots, \mathbf{r}_m\}, \tag{1}$$

where each ray $\mathbf{r}_i \in \mathbb{R}^6$ is associated with a known pixel coordinate $\mathbf{u}_i$. We parameterize each ray $\mathbf{r}$ traveling in direction $\mathbf{d} \in \mathbb{R}^3$ through any point $\mathbf{p} \in \mathbb{R}^3$ using Plücker coordinates (Plücker, 1828):

$$\mathbf{r} = \langle \mathbf{d}, \mathbf{m} \rangle \in \mathbb{R}^6, \tag{2}$$

where $\mathbf{m} = \mathbf{p} \times \mathbf{d} \in \mathbb{R}^3$ is the moment vector, and importantly, is agnostic to the specific point on the ray used to compute it. When $\mathbf{d}$ is of unit length, the norm of the moment $\mathbf{m}$ represents the distance from the ray to the origin.

**Converting from Camera to Ray Bundle.** Given a known camera and a set of 2D pixel coordinates $\{\mathbf{u}_i\}_m$, the directions $\mathbf{d}$ can be computed by unprojecting rays from the pixel coordinates, and the moments $\mathbf{m}$ can be computed by treating the camera center as the point $\mathbf{p}$ since all rays intersect at the camera center:

$$\mathbf{d} = \mathbf{R}^\top \mathbf{K}^{-1} \mathbf{u}, \qquad \mathbf{m} = (-\mathbf{R}^\top \mathbf{t}) \times \mathbf{d}. \tag{3}$$

In practice, we select the points $\{\mathbf{u}_i\}_m$ by uniformly sampling points on a grid across the image or image crop, as shown in Fig. 2. This allows us to associate each patch in the image with a ray passing through the center of the patch, which we will use later to design a patch- and ray-conditioned architecture.

**Converting from Ray Bundle to Camera.** Given a collection of rays $\mathcal{R} = \{\mathbf{r}_i\}_m$ associated with 2D pixels $\{\mathbf{u}_i\}_m$, we show that one can recover the camera extrinsics and intrinsics. We start by solving for the camera center $\mathbf{c}$ by finding the 3D world coordinate closest to the intersection of all rays in $\mathcal{R}$:

$$\mathbf{c} = \arg\min_{\mathbf{p} \in \mathbb{R}^3} \sum_{\langle \mathbf{d}, \mathbf{m} \rangle \in \mathcal{R}} \|\mathbf{p} \times \mathbf{d} - \mathbf{m}\|^2. \tag{4}$$

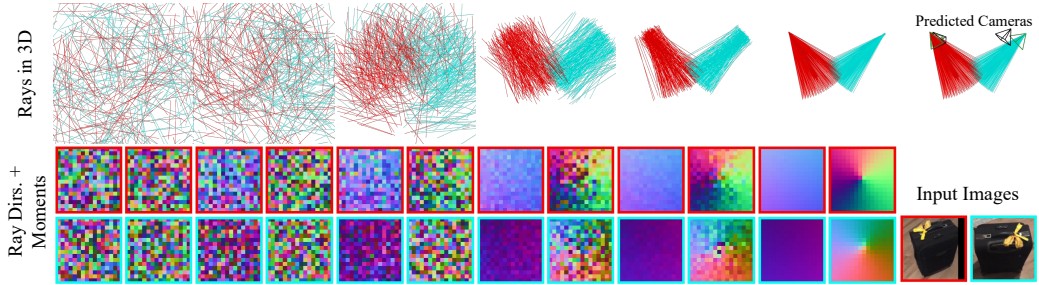

Figure 4: **Visualizing the Denoising Process Using Our Ray Diffuser.** Given the 2 images of the suitcase (*Bottom Right*), we visualize the denoising process starting from randomly initialized camera rays. We visualize the noisy rays using the Plücker representation (ray directions and moments) in the bottom row and their corresponding 3D positions in the top row. In the rightmost column, we recover the predicted cameras (green) and compare them to the ground truth cameras (black).

To solve for the rotation $R$ (and intrinsics $K$) for each camera, we can solve for the optimal homography matrix $P$ that transforms per-pixel ray directions from the predicted ones to those of an 'identity' camera ($K = I$ and $R = I$):

$$P = \arg\min_{\|H\|=1} \sum_{i=1}^{m} \|H\mathbf{d}_i \times \mathbf{u}_i\| . \tag{5}$$

The matrix $P$ can be computed via DLT (Abdel-Aziz et al., 2015) and can allow recovering $R$ using RQ-decomposition as $K$ is an upper-triangular matrix and $R$ is orthonormal. Once the camera rotation $R$ and camera center $c$ are recovered, the translation $t$ can be computed as $t = -Rc$.

## 3.2 POSE ESTIMATION VIA RAY REGRESSION

We now describe an approach for predicting the ray representation outlined in Sec. 3.1 for camera pose estimation given $N$ images $\{I_1, \ldots, I_N\}$. Given ground truth camera parameters, we can compute the ground truth ray bundles $\{\mathcal{R}_1, \ldots, \mathcal{R}_N\}$. As shown in Fig. 2, we compute the rays over a uniform $p \times p$ grid over the image such that each ray bundle consists of $m = p^2$ rays (eq. (1)).

To ensure a correspondence between rays and image patches, we use a spatial image feature extractor and treat each patch feature as a token:

$$f_{\text{feat}}(I) = \boldsymbol{f} \in \mathbb{R}^{p \times p \times d} . \tag{6}$$

To make use of the crop parameters, we also concatenate the pixel coordinate $\boldsymbol{u}$ (in normalized device coordinates with respect to the uncropped image) to each spatial feature. We use a transformer-based architecture (Dosovitskiy et al. (2021); Peebles & Xie (2023)) that jointly processes each of the $p^2$ tokens from $N$ images, and predicts the ray corresponding to each patch:

$$\{\hat{\mathcal{R}}\}_{i=1}^{N} = f_{\text{Regress}} \left( \{\boldsymbol{f}_i, \boldsymbol{u}_i\}_{i=1}^{N \cdot p^2} \right) . \tag{7}$$

We train the network by computing a reconstruction loss on the predicted camera rays:

$$\mathcal{L}_{\text{recon}} = \sum_{i=1}^{N} \left\| \hat{\mathcal{R}}_i - \mathcal{R}_i \right\|_2^2 . \tag{8}$$

## 3.3 POSE ESTIMATION VIA DENOISING RAY DIFFUSION

While the patchwise regression-based architecture described in Sec. 3.2 can effectively predict our distributed ray-based parametrization, the task of predicting poses (in the form of rays) may still be ambiguous given sparse views. To handle inherent uncertainty in the predictions (due to symmetries and partial observations), we extend the previously described regression approach to instead learn a diffusion-based probabilistic model over our distributed ray representation.

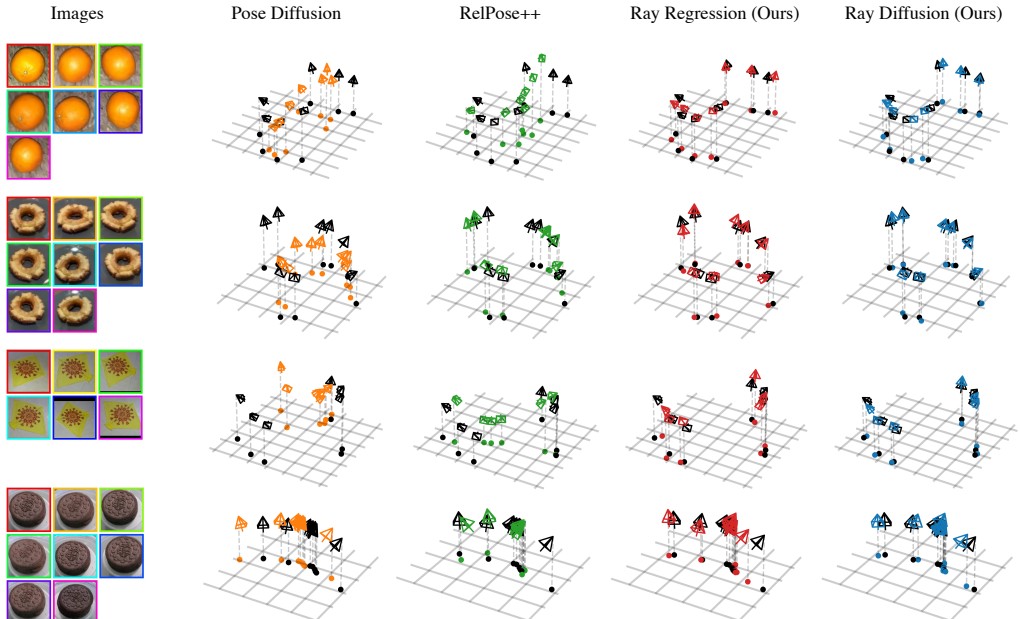

Figure 5: **Qualitative Comparison Between Predicted Camera Poses.** We compare the results of our regression and diffusion approaches with PoseDiffusion and RelPose++. Ground truth (black) camera trajectories are aligned to the predicted (colored) camera trajectories by performing Procrustes optimal alignment on the camera centers. The top two examples are from seen categories, and the bottom two are from held out categories.

Denoising diffusion models (Ho et al., 2020) approximate a data likelihood function by inverting a noising process that adds time-dependent Gaussian noise to the original sample $x_0$:

$$x_t = \sqrt{\bar{\alpha}_t}x_0 + \sqrt{1 - \bar{\alpha}_t}\epsilon, \tag{9}$$

where $\epsilon \sim \mathcal{N}(0, \boldsymbol{I})$ and $\alpha_t$ is a hyper-parameter schedule of noise weights such that $x_T$ can be approximated as a standard Gaussian distribution. To learn the reverse process, one can train a denoising network $f_\theta$ to predict the denoised sample $\boldsymbol{x}_0$ conditioned on $\boldsymbol{x}_t$:

$$\mathcal{L}(\theta) = \mathbb{E}_{t,\boldsymbol{x}_0,\epsilon} \left\| x_0 - f_\theta\left(x_t, t\right) \right\|^2. \tag{10}$$

We instantiate this denoising diffusion framework to model the distributions over patchwise rays conditioned on the input images. We do this by simply modifying our ray regression network from Sec. 3.2 to be additionally conditioned on noisy rays (concatenated with patchwise features and pixel coordinates) and a positionally encoded (Vaswani et al., 2017) time embedding $t$:

$$\{\hat{\mathcal{R}}\}_{i=1}^N = f_{\text{Diffuse}}\left(\{(\boldsymbol{f}_i, \boldsymbol{u}_i, \boldsymbol{r}_{i,t})\}_{i=1}^{N \cdot p^2}, t\right), \tag{11}$$

where the noisy rays $\boldsymbol{r}_{i,t}$ can be computed as:

$$\boldsymbol{r}_{i,t} = \sqrt{\bar{\alpha}_t}\boldsymbol{r}_i + \sqrt{1 - \bar{\alpha}_t}\epsilon. \tag{12}$$

Conveniently, our time-conditioned ray denoiser can be trained with the same L2 loss function (eq. (8)) as our ray regressor. We visualize the states of the denoised rays during backward diffusion in Fig. 4.

### 3.4 IMPLEMENTATION DETAILS

Following Lin et al. (2024), we place the world origin at the point closest to the optical axes of the training cameras, which represents a useful inductive bias for center-facing camera setups. To handle coordinate system ambiguity, we rotate the world coordinates such that the first camera always has identity rotation and re-scale the scene such that the first camera translation has unit norm. Following

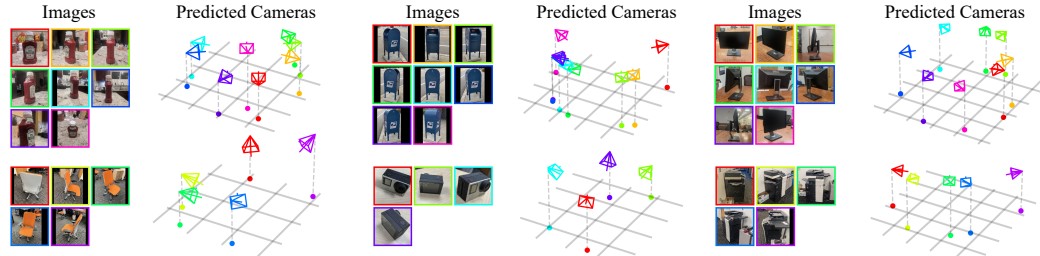

Figure 6: **Generalization to In-the-wild Self-captures.** We test the generalization of our ray diffusion model on a variety of *self-captured data* on objects that are not in CO3D.

prior work (Zhang et al., 2022), we take square image crops tightly around the object bounding box and adjust the uniform grid of pixel coordinates associated with the rays accordingly.

We use a pre-trained, frozen DINOv2 (S/14) (Oquab et al., 2023) as our image feature extractor. We use a DiT (Peebles & Xie, 2023) with 16 transformer blocks as the architecture for both $f_{\text{Regress}}$ (with $t$ always set to 100) and $f_{\text{Diffusion}}$. We train our diffusion model with $T=100$ timesteps. When training our denoiser, we add noise to the direction and moment representation of rays. The ray regression and ray diffusion models take about 2 and 4 days respectively to train on 8 A6000 GPUs.

To predict cameras with our ray denoiser, we use DDPM (Ho et al., 2020) inference with slight modifications. Empirically, we found that removing the stochastic noise in DDPM inference and stopping the backward diffusion process early (and using the predicted $x_0$ as the estimate) produced better performance. We hypothesize that this is because while the earlier diffusion steps help select among distinct plausible modes, the later steps yield samples around these—and this may be detrimental to accuracy metrics that prefer distribution modes.

# 4 EXPERIMENTS

## 4.1 EXPERIMENTAL SETUP

**Dataset.** Our method is trained and evaluated using CO3Dv2 (Reizenstein et al., 2021). This dataset contains turntable videos spanning 51 categories of household objects. Each frame is labeled with poses determined by COLMAP (Schönberger et al., 2016; Schönberger & Frahm, 2016). Following Zhang et al. (2022), we train on 41 categories and hold out the remaining 10 categories for evaluating generalization.

**Baselines.** We evaluate our method against a handful of learning-based and correspondence-based pose estimation works.

*COLMAP (Schönberger et al., 2016; Schönberger & Frahm, 2016).* COLMAP is a standard dense correspondence-based SfM pipeline. We use an implementation (Sarlin et al., 2019) which uses SuperPoint features (DeTone et al., 2018) and SuperGlue matching (Sarlin et al., 2020).

*RelPose (Zhang et al., 2022).* RelPose predicts relative rotations between pairs of cameras and defines evaluation procedures to optimize over a learned scoring function and determine joint rotations.

*RelPose++ (Lin et al., 2024).* RelPose++ builds upon the pairwise scoring network of RelPose to incorporate multi-view reasoning via a transformer and also allows predicting camera translations.

*R+T Regression (Lin et al., 2024).* To test the importance of modeling uncertainty, Lin et al. (2024) trains a baseline that directly regresses poses. We report the numbers from Lin et al. (2024).

*PoseDiffusion (Wang et al., 2023).* PoseDiffusion reformulates the pose estimation task as directly diffusing camera extrinsics and focal length. Additionally, they introduce a geometry-guided sampling error to enforce epipolar constraints on predicted poses. We evaluate PoseDiffusion with and without the geometry-guided sampling.

| | # of Images | 2 | 3 | 4 | 5 | 6 | 7 | 8 |
|---|---|---|---|---|---|---|---|---|
| Seen Categories | COLMAP (SP+SG) (Sarlin et al., 2019) | 30.7 | 28.4 | 26.5 | 26.8 | 27.0 | 28.1 | 30.6 |
| | RelPose (Zhang et al., 2022) | 56.0 | 56.5 | 57.0 | 57.2 | 57.2 | 57.3 | 57.2 |
| | PoseDiffusion w/o GGS (Wang et al., 2023) | 74.5 | 75.4 | 75.6 | 75.7 | 76.0 | 76.3 | 76.5 |
| | PoseDiffusion (Wang et al., 2023) | 75.7 | 76.4 | 76.8 | 77.4 | 78.0 | 78.7 | 78.8 |
| | RelPose++ (Lin et al., 2024) | 81.8 | 82.8 | 84.1 | 84.7 | 84.9 | 85.3 | 85.5 |
| | R+T Regression (Lin et al., 2024) | 49.1 | 50.7 | 53.0 | 54.6 | 55.7 | 56.1 | 56.5 |
| | Ray Regression (Ours) | 88.8 | 88.7 | 88.7 | 89.0 | 89.4 | 89.3 | 89.2 |
| | Ray Diffusion (Ours) | **91.8** | **92.4** | **92.6** | **92.9** | **93.1** | **93.3** | **93.3** |
| Unseen Categories | COLMAP (SP+SG) (Sarlin et al., 2019) | 34.5 | 31.8 | 31.0 | 31.7 | 32.7 | 35.0 | 38.5 |
| | RelPose (Zhang et al., 2022) | 48.6 | 47.5 | 48.1 | 48.3 | 48.4 | 48.4 | 48.3 |
| | PoseDiffusion w/o GGS (Wang et al., 2023) | 62.1 | 62.4 | 63.0 | 63.5 | 64.2 | 64.2 | 64.4 |
| | PoseDiffusion (Wang et al., 2023) | 63.2 | 64.2 | 64.2 | 65.7 | 66.2 | 67.0 | 67.7 |
| | RelPose++ (Lin et al., 2024) | 69.8 | 71.1 | 71.9 | 72.8 | 73.8 | 74.4 | 74.9 |
| | R+T Regression (Lin et al., 2024) | 42.7 | 43.8 | 46.3 | 47.7 | 48.4 | 48.9 | 48.9 |
| | Ray Regression (Ours) | 79.0 | 79.6 | 80.6 | 81.4 | 81.3 | 81.9 | 81.9 |
| | Ray Diffusion (Ours) | **83.5** | **85.6** | **86.3** | **86.9** | **87.2** | **87.5** | **88.1** |

Table 1: **Camera Rotation Accuracy on CO3D (@ 15°).** Here we report the proportion of relative camera rotations that are within 15 degrees of the ground truth.

| | # of Images | 2 | 3 | 4 | 5 | 6 | 7 | 8 |
|---|---|---|---|---|---|---|---|---|
| Seen Categories | COLMAP (SP+SG) (Sarlin et al., 2019) | 100 | 34.5 | 23.8 | 18.9 | 15.6 | 14.5 | 15.0 |
| | PoseDiffusion w/o GGS (Wang et al., 2023) | 100 | 76.5 | 66.9 | 62.4 | 59.4 | 58.0 | 56.5 |
| | PoseDiffusion (Wang et al., 2023) | 100 | 77.5 | 69.7 | 65.9 | 63.7 | 62.8 | 61.9 |
| | RelPose++ (Lin et al., 2024) | 100 | 85.0 | 78.0 | 74.2 | 71.9 | 70.3 | 68.8 |
| | R+T Regression (Lin et al., 2024) | 100 | 58.3 | 41.6 | 35.9 | 32.7 | 31.0 | 30.0 |
| | Ray Regression (Ours) | 100 | 91.7 | 85.7 | 82.1 | 79.8 | 77.9 | 76.2 |
| | Ray Diffusion (Ours) | 100 | **94.2** | **90.5** | **87.8** | **86.2** | **85.0** | **84.1** |
| Unseen Categs. | COLMAP (SP+SG) (Sarlin et al., 2019) | 100 | 36.0 | 25.5 | 20.0 | 17.9 | 17.6 | 19.1 |
| | PoseDiffusion w/o GGS(Wang et al., 2023) | 100 | 62.5 | 48.8 | 41.9 | 39.0 | 36.5 | 34.8 |
| | PoseDiffusion (Wang et al., 2023) | 100 | 63.6 | 50.5 | 45.7 | 43.0 | 41.2 | 39.9 |
| | RelPose++ (Lin et al., 2024) | 100 | 70.6 | 58.8 | 53.4 | 50.4 | 47.8 | 46.6 |
| | R+T Regression (Lin et al., 2024) | 100 | 48.9 | 32.6 | 25.9 | 23.7 | 22.4 | 21.3 |
| | Ray Regression (Ours) | 100 | 83.7 | 75.6 | 70.8 | 67.4 | 65.3 | 63.9 |
| | Ray Diffusion (Ours) | 100 | **87.7** | **81.1** | **77.0** | **74.1** | **72.4** | **71.4** |

Table 2: **Camera Center Accuracy on CO3D (@ 0.1).** Here we report the proportion of camera centers that are within 0.1 of the scene scale. We apply an optimal similarity transform $(s, \boldsymbol{R}, \boldsymbol{t})$ to align predicted camera centers to ground truth camera centers (hence the alignment is perfect at $N = 2$ but worsens with more images).

## 4.2 METRICS

We evaluate sparse image sets of 2 to 8 images for each test sequence in CO3D. For an $N$ image evaluation, we randomly sample $N$ images and compute the accuracy of the predicted poses. We average these accuracies over 5 samples for each sequence to reduce stochasticity.

*Rotation Accuracy.* We first compute the relative rotations between each pair of cameras for both predicted and ground truth poses. Then we determine the error between the ground truth and predicted pairwise relative rotations and report the proportion of these errors within 15 degrees.

*Camera Center Accuracy.* We align the ground truth and predicted poses in CO3D using the optimal similarity transform $(s, \boldsymbol{R}, \boldsymbol{t})$. We compare our prediction to the scene scale (the distance from the scene centroid to the farthest camera, following Sinha et al. (2023)). We report the proportion of aligned camera centers within 10 percent of the scene scale to the ground truth.

## 4.3 EVALUATION

We report the camera rotation accuracy in Tab. 1 and camera center accuracy in Tab. 2 evaluated on CO3D. We find that COLMAP struggles in wide-baseline settings due to insufficient image overlap

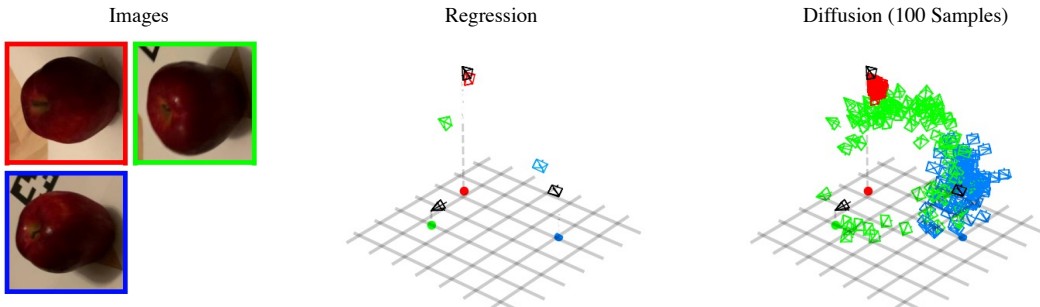

Figure 7: **Modeling Uncertainty Via Sampling Modes.** Sparse-view camera poses are sometimes inherently ambiguous due to symmetry. The capacity to model such uncertainty in probabilistic models such as our Ray Diffusion model is a significant advantage over regression-based models that must commit to a single mode. We thus investigate taking multiple samples from our diffusion model. We visualize the predicted cameras (colored) of both our regression- and diffusion-based approaches compared to the ground truth (black). While the regression model predicts the green camera incorrectly, we can recover better modes by sampling our diffusion model multiple times.

to find correspondences. We find that both the regression and diffusion versions of our method safely outperform all existing approaches, suggesting that our ray-based representation can effectively recover precise camera poses in this setup. In particular, our ray regression method significantly outperforms the baseline that regresses extrinsics R and T directly (R+T Regression). Similarly, our ray diffusion model demonstrates a large improvement over R+T Diffusion (PoseDiffusion without GGS) (Wang et al., 2023), while also outperforming their full method (PoseDiffusion) which includes geometry-guided sampling.

We show qualitative results comparing both our Ray Regression and Diffusion methods with PoseDiffusion and RelPose++ in Fig. 5. We find that our ray-based representation consistently achieves finer localization. Additionally, ray diffusion achieves slightly better performance than ray regression. More importantly, it also allows recovering multiple plausible modes under uncertainty, as highlighted in Fig. 7.

**Ablating Ray Resolution.** We conduct an ablation study to evaluate how the number of camera rays affects performance in Tab. 3. We find that increasing the number of camera rays significantly improves performance. Note that we kept the parameter count of the transformer constant, but more tokens incur a greater computational cost. All other experiments are conducted with $16 \times 16$ rays.

| # of Rays | Rot@15 | CC@0.01 |
|-----------|--------|---------|
| $2 \times 2$ | 52.5 | 72.5 |
| $4 \times 4$ | 70.3 | 82.6 |
| $8 \times 8$ | 76.1 | 84.8 |
| $16 \times 16$ | **84.0** | **89.8** |

Table 3: **Ray Resolution Ablation.** We evaluate various numbers of patches/rays by training a category-specific model for 2 different training categories (hydrant, wineglass) with $N = 3$ images. Performance across the 2 categories is averaged. We find that increasing the number of rays significantly improves performance. However, we found that increasing the number of rays beyond $16 \times 16$ was computationally prohibitive.

**Demonstration on Self-captures.** Finally, to demonstrate that our approach generalizes beyond the distribution of sequences from CO3D, we show qualitative results using Ray Diffusion on a variety of in-the-wild self-captures in Fig. 6.

## 5 DISCUSSION

In this work, we explored representing camera poses using a distributed ray representation, and proposed a deterministic regression and a probabilistic diffusion approach for predicting these rays. While we examined this representation in the context of sparse views, it can be explored for single-view or dense multi-view setups. In addition, while our representation allows implicitly leveraging associations between patches, we do not enforce any geometric consistency (as done in classical pose estimation pipelines). It would be interesting to explore joint inference of our distributed ray representation and geometry in future work.

**Acknowledgements.** We would like to thank Shikhar Bahl and Samarth Sinha for their feedback on drafts. This work was supported in part by the NSF GFRP (Grant No. DGE1745016), a CISCO

gift award, and the Intelligence Advanced Research Projects Activity (IARPA) via Department of Interior/Interior Business Center (DOI/IBC) contract number 140D0423C0074. The U.S. Government is authorized to reproduce and distribute reprints for Governmental purposes notwithstanding any copyright annotation thereon. Disclaimer: The views and conclusions contained herein are those of the authors and should not be interpreted as necessarily representing the official policies or endorsements, either expressed or implied, of IARPA, DOI/IBC, or the U.S. Government.

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

# A APPENDIX

In this section, we include the following:

- Additional qualitative comparisons on both seen (Fig. 8) and unseen (Fig. 9) object categories.
- Evaluations on CO3D at multiple thresholds (Tabs. 10 and 11).
- Evaluations on CO3D using Area-under-Curve (AUC) to account for all thresholds (Tabs. 2 and 8 and Fig. 11).
- Evaluation of inference time of all methods (Tab. 6).
- Benchmark of memory usage of our diffusion model (Tab. 7).
- Generalization of our method on up to 43 images (trained on 8 images) in Tab. 5 and to RealEstate10K (Zhou et al., 2018) (trained on CO3D).
- Qualitative results on training on SfM-style datasets (MegaDepth (Li & Snavely, 2018)) in Fig. 10.

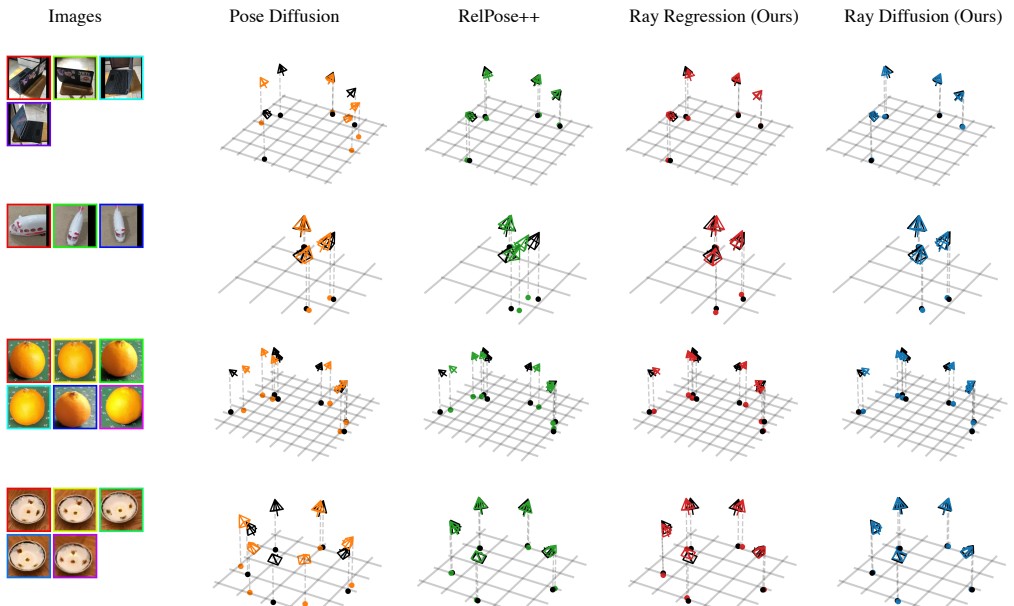

Figure 8: **Additional Qualitative Results for Seen Categories.**

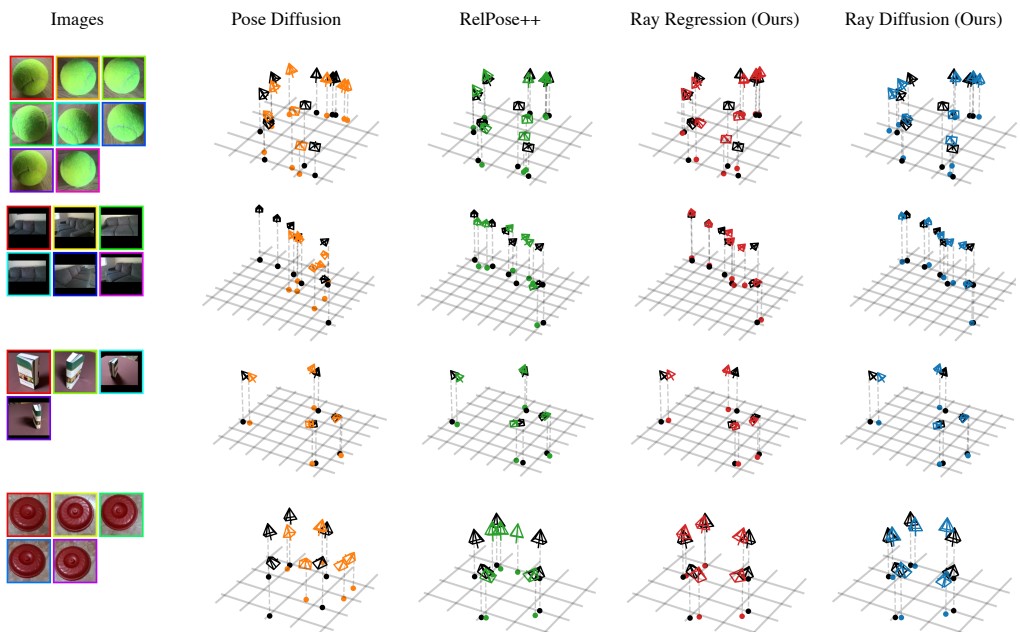

Figure 9: **Additional Qualitative Results for Unseen Categories.**

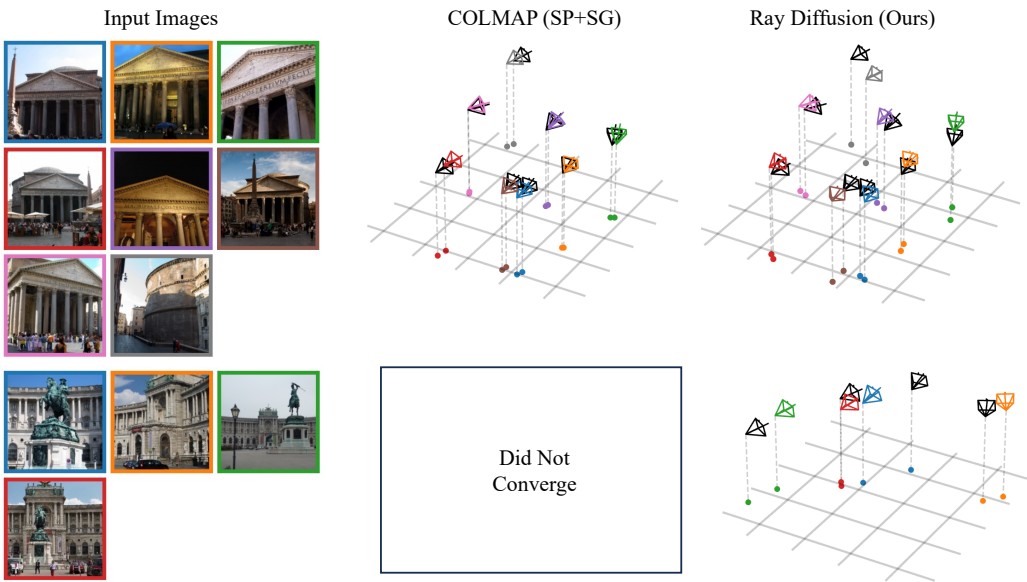

Figure 10: **Qualitative Results on MegaDepth Dataset (Li & Snavely, 2018).** As a proof of concept on scene-level datasets, we train our ray diffusion model on 235 of the SfM reconstructions from MegaDepth. At training time, we normalize the scene such that the first camera has identity rotation and zero translation and rescale the scene such that the standard deviation in the ground truth ray moments is constant. Here we visualize predicted camera poses in color and ground truth cameras in black for held-out scenes from the dataset. The color of the camera (and color of the circle for ground truth cameras) indicate the correspondence with the input image. We compare our method with COLMAP (with SuperPoint+SuperGlue) which is highly accurate when it converges. Our method is more robust but less accurate when COLMAP does converge.

| | # of Images | 2 | 3 | 4 | 5 | 6 | 7 | 8 |
|---|---|---|---|---|---|---|---|---|
| **Rot. Acc.** | Constant Rot. | 84.0 | 83.8 | 83.9 | 84.0 | 84.0 | 84.0 | 83.9 |
| | PoseDiffusion | 77.6 | 77.9 | 78.4 | 78.7 | 78.9 | 79.3 | 79.0 |
| | RelPose++ | 83.8 | 85.1 | 85.8 | 86.4 | 86.5 | 86.7 | 86.8 |
| | Ray Regression (Ours) | 90.8 | **90.0** | 89.9 | **89.7** | **89.5** | **89.5** | **89.5** |
| | Ray Diffusion (Ours) | **90.9** | 89.9 | 89.5 | 89.3 | 89.1 | 88.8 | 88.3 |
| **CC. Acc.** | PoseDiffusion | 100 | 77.7 | 65.9 | 60.1 | 55.0 | 52.2 | 50.2 |
| | RelPose++ | 100 | 71.2 | 60.6 | 54.0 | 49.4 | 47.1 | 45.5 |
| | Ray Regression (Ours) | 100 | 74.4 | 62.0 | 56.0 | 51.3 | 49.2 | 47.1 |
| | Ray Diffusion (Ours) | 100 | **79.7** | **68.6** | **62.2** | **57.8** | **54.9** | **52.1** |

Table 4: **Evaluation of Rotation and Camera Center Accuracy on RealEstate10K (Zhou et al., 2018).** Here we report zero-shot generalization of methods trained on CO3D and tested on RealEstate10K without any fine-tuning. We measure rotation accuracy at a threshold of 15 degrees and camera center accuracy at a threshold of 0.2. The constant rotation baseline always predicts an identity rotation. We find that this dataset has a strong, forward-facing bias, so even naively predicting an identity rotation performs well.

| # of Images | 8 | 15 | 22 | 29 | 36 | 43 |
|---|---|---|---|---|---|---|
| Rotation Acc. (Seen Categories) | 93.3 | 93.1 | 93.3 | 93.1 | 93.4 | 93.4 |
| Rotation Acc. (Unseen Categories) | 88.1 | 88.2 | 89.2 | 88.7 | 89.0 | 88.9 |
| Camera Center Acc. (Seen Categories) | 84.1 | 78.3 | 76.5 | 75.3 | 74.7 | 74.2 |
| Camera Center Acc. (Unseen Categories) | 71.4 | 62.7 | 61.1 | 59.3 | 59.2 | 58.9 |

Table 5: **Generalization to More Images on CO3D using Ray Diffusion.** Our ray diffusion model is trained with between 2 and 8 images. At inference time, we find that we can effectively run backward diffusion with more images by randomly sampling new mini-batches at each iteration of DDPM (keeping the first image fixed).

| | Inference Time (s) |
|---|---|
| COLMAP (SP+SG) (Sarlin et al., 2019) | 2.06 |
| RelPose (Zhang et al., 2022) | 29.5 |
| PoseDiffusion w/o GGS (Wang et al., 2023) | 0.304 |
| PoseDiffusion (Wang et al., 2023) | 2.83 |
| RelPose++ (Lin et al., 2024) | 4.49 |
| R+T Regression (Lin et al., 2024) | 0.0300 |
| Ray Regression (Ours) | 0.133 |
| Ray Diffusion (Ours) | 11.1 |

Table 6: **Inference time for N=8 Images.** All benchmarks are completed using a single Nvidia A6000 GPU. We compute the best of 5 runs. Ray Diffusion and PoseDiffusion both use DDPM inference with 100 steps. RelPose uses 200 iterations of coordinate ascent while RelPose++ uses 50 iterations. Unsurprisingly, we find that feedforward methods (Ray Regression, R+T Regression) achieve very low latency. The other methods require lengthier optimization loops.

| # of Images | 2 | 3 | 4 | 5 | 6 | 7 | 8 |
|---|---|---|---|---|---|---|---|
| Memory Usage (MiB) | 2877 | 2903 | 2913 | 2965 | 3007 | 3013 | 3095 |

Table 7: **Memory Usage of our Ray Diffusion Model.** We measure the GPU memory usage when running inference using our Ray Diffusion model with various numbers of images. We report the peak memory consumed as reported by `nvidia-smi`, which may not be exact. Note that loading our model into memory consumes 2637 MiB, which is the majority of the memory usage. We observe sub-quadratic growth in memory usage, likely because the DINO feature computation is heavier than the ray transformer.

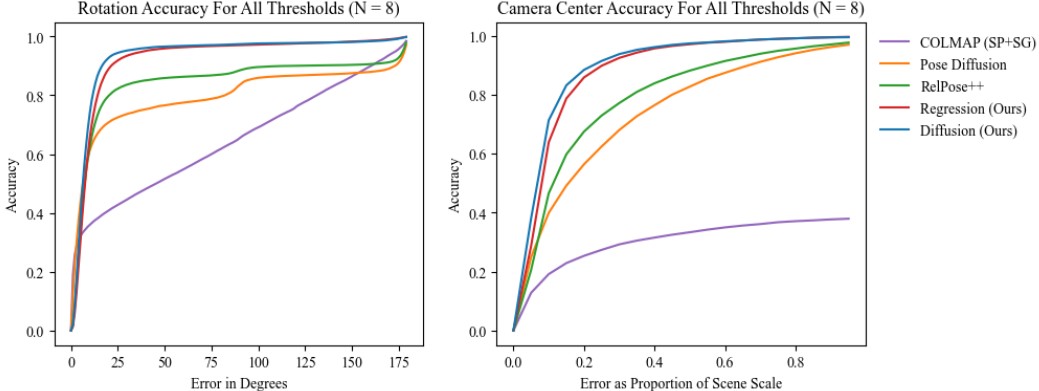

Figure 11: **Accuracy For All Thresholds** We visualize our unseen category accuracy curves for 8 images. Note for very fine thresholds of rotation accuracy, between 1 and 5 degrees, COLMAP rises faster than all other methods. This is due to the precision of COLMAP when it is able to converge to a reasonable set of poses. However, COLMAP is only able to converge in roughly 40% of our test time evaluations. See Tabs. 8 and 9 for AUC metrics for all numbers of images.

| | # of Images | 2 | 3 | 4 | 5 | 6 | 7 | 8 |
|---|---|---|---|---|---|---|---|---|
| Seen Cate. | COLMAP (SP+SG) (Sarlin et al., 2019) | 57.8 | 56.9 | 56.8 | 57.5 | 58.1 | 58.9 | 60.3 |
| | PoseDiffusion (Wang et al., 2023) | 84.9 | 84.9 | 85.3 | 85.6 | 85.9 | 86.2 | 86.5 |
| | RelPose++ (Lin et al., 2024) | 88.7 | 89.2 | 89.8 | 90.3 | 90.4 | 90.8 | 90.7 |
| | Ray Regression (Ours) | 93.6 | 93.4 | 93.3 | 93.5 | 93.6 | 93.6 | 93.5 |
| | Ray Diffusion (Ours) | **94.3** | **94.4** | **94.4** | **94.5** | **94.6** | **94.6** | **94.7** |
| Unseen Cate. | COLMAP (SP+SG) (Sarlin et al., 2019) | 59.1 | 58.8 | 59.1 | 60.2 | 61.2 | 62.6 | 64.6 |
| | PoseDiffusion (Wang et al., 2023) | 76.8 | 77.0 | 77.3 | 77.8 | 78.3 | 78.5 | 79.0 |
| | RelPose++ (Lin et al., 2024) | 79.4 | 81.2 | 81.7 | 82.8 | 83.4 | 83.6 | 83.8 |
| | Ray Regression (Ours) | 91.2 | 91.0 | 91.5 | 91.5 | 91.5 | 91.7 | 91.7 |
| | Ray Diffusion (Ours) | **91.7** | **92.2** | **92.5** | **92.9** | **92.6** | **92.8** | **92.8** |

Table 8: **Rotation Accuracy AUC on CO3D.** Here we report the rotation accuracy across all thresholds (0 to 180 degrees in 1 degree increments) by measuring the Area Under Curve (AUC). See Fig. 11 for a visualization of the curves for N=8.

| | # of Images | 2 | 3 | 4 | 5 | 6 | 7 | 8 |
|---|---|---|---|---|---|---|---|---|
| Seen Cate. | COLMAP (SP+SG) (Sarlin et al., 2019) | 40.1 | 34.1 | 25.9 | 22.4 | 20.6 | 20.5 | 22.0 |
| | PoseDiffusion (Wang et al., 2023) | 92.5 | 85.6 | 82.3 | 80.4 | 79.4 | 78.9 | 78.6 |
| | RelPose++ (Lin et al., 2024) | 92.5 | 85.6 | 82.3 | 80.4 | 79.4 | 78.9 | 78.6 |
| | Ray Regression (Ours) | 92.5 | 90.2 | 88.5 | 87.6 | 87.1 | 86.6 | 86.2 |
| | Ray Diffusion (Ours) | 92.5 | **90.7** | **89.5** | **88.8** | **88.3** | **88.0** | **87.8** |
| Unseen Cate. | COLMAP (SP+SG) (Sarlin et al., 2019) | 41.0 | 36.6 | 29.4 | 25.8 | 25.0 | 26.2 | 28.7 |
| | PoseDiffusion (Wang et al., 2023) | 92.5 | 81.3 | 75.8 | 73.3 | 71.4 | 70.3 | 69.7 |
| | RelPose++ (Lin et al., 2024) | 92.5 | 83.3 | 79.0 | 76.9 | 75.7 | 74.8 | 74.3 |
| | Ray Regression (Ours) | 92.5 | 88.6 | 86.3 | 85.0 | 84.1 | 83.6 | 83.2 |
| | Ray Diffusion (Ours) | 92.5 | **89.2** | **87.2** | **86.2** | **85.4** | **84.9** | **84.6** |

Table 9: **Camera Center Accuracy AUC on CO3D.** Here we report the camera center accuracy across all thresholds (0 to 1 of scene scale in 0.05 increments) by measuring the Area Under Curve (AUC). See Fig. 11 for a visualization of the curves for N=8.

| | # of Images | 2 | 3 | 4 | 5 | 6 | 7 | 8 |
|---|---|---|---|---|---|---|---|---|
| Seen Categories | Ray Regression @ 5 | 39.8 | 38.5 | 38.7 | 38.7 | 38.7 | 38.6 | 38.4 |
| | Ray Regression @ 10 | 75.3 | 75.4 | 75.5 | 76.0 | 76.2 | 76.1 | 76.1 |
| | Ray Regression @ 15 | 88.8 | 88.7 | 88.7 | 89.0 | 89.4 | 89.3 | 89.2 |
| | Ray Regression @ 30 | 96.0 | 95.8 | 95.7 | 95.8 | 96.0 | 96.0 | 95.8 |
| | Ray Diffusion @ 5 | 48.7 | 48.5 | 48.6 | 49.0 | 49.2 | 49.2 | 49.3 |
| | Ray Diffusion @ 10 | 81.9 | 82.4 | 83.0 | 83.4 | 83.7 | 84.0 | 84.2 |
| | Ray Diffusion @ 15 | 91.8 | 92.4 | 92.6 | 92.9 | 93.1 | 93.3 | 93.3 |
| | Ray Diffusion @ 30 | 96.7 | 96.9 | 96.9 | 97.1 | 97.2 | 97.2 | 97.3 |
| Unseen Categories | Ray Regression @ 5 | 29.2 | 29.3 | 29.9 | 29.9 | 30.6 | 29.9 | 30.1 |
| | Ray Regression @ 10 | 61.8 | 63.3 | 64.8 | 64.9 | 65.4 | 65.7 | 65.4 |
| | Ray Regression @ 15 | 79.0 | 79.6 | 80.6 | 81.4 | 81.3 | 81.9 | 81.9 |
| | Ray Regression @ 30 | 92.8 | 92.4 | 93.0 | 93.3 | 93.0 | 93.4 | 93.5 |
| | Ray Diffusion @ 5 | 37.7 | 36.6 | 37.8 | 38.2 | 38.6 | 38.2 | 38.6 |
| | Ray Diffusion @ 10 | 68.5 | 70.3 | 71.4 | 72.3 | 73.2 | 73.1 | 73.8 |
| | Ray Diffusion @ 15 | 83.5 | 85.6 | 86.3 | 86.9 | 87.2 | 87.5 | 88.1 |
| | Ray Diffusion @ 30 | 93.8 | 94.4 | 94.8 | 95.4 | 95.0 | 95.2 | 95.2 |

Table 10: **Camera Rotation Accuracy on CO3D at varying thresholds.**

| | # of Images | 2 | 3 | 4 | 5 | 6 | 7 | 8 |
|---|---|---|---|---|---|---|---|---|
| Seen Cate. | Ray Regression @ 0.05 | 100.0 | 74.3 | 57.7 | 49.6 | 45.0 | 41.7 | 39.2 |
| | Ray Regression @ 0.1 | 100.0 | 91.7 | 85.7 | 82.1 | 79.8 | 77.9 | 76.2 |
| | Ray Regression @ 0.2 | 100.0 | 97.4 | 95.5 | 94.3 | 93.9 | 93.2 | 92.7 |
| | Ray Diffusion @ 0.05 | 100.0 | 80.8 | 67.8 | 60.5 | 56.2 | 53.5 | 51.1 |
| | Ray Diffusion @ 0.1 | 100.0 | 94.2 | 90.5 | 87.8 | 86.2 | 85.0 | 84.1 |
| | Ray Diffusion @ 0.2 | 100.0 | 98.0 | 96.6 | 96.0 | 95.5 | 95.1 | 94.8 |
| Unseen Cate. | Ray Regression @ 0.05 | 100.0 | 64.7 | 46.8 | 38.0 | 33.4 | 29.9 | 28.6 |
| | Ray Regression @ 0.1 | 100.0 | 83.7 | 75.6 | 70.8 | 67.4 | 65.3 | 63.9 |
| | Ray Regression @ 0.2 | 100.0 | 94.3 | 90.8 | 88.6 | 87.4 | 86.8 | 85.9 |
| | Ray Diffusion @ 0.05 | 100.0 | 70.6 | 54.7 | 47.5 | 42.3 | 39.0 | 38.0 |
| | Ray Diffusion @ 0.1 | 100.0 | 87.7 | 81.1 | 77.0 | 74.1 | 72.4 | 71.4 |
| | Ray Diffusion @ 0.2 | 100.0 | 94.7 | 92.2 | 90.9 | 89.8 | 89.2 | 88.5 |

Table 11: **Camera Center Accuracy on CO3D at varying thresholds.**

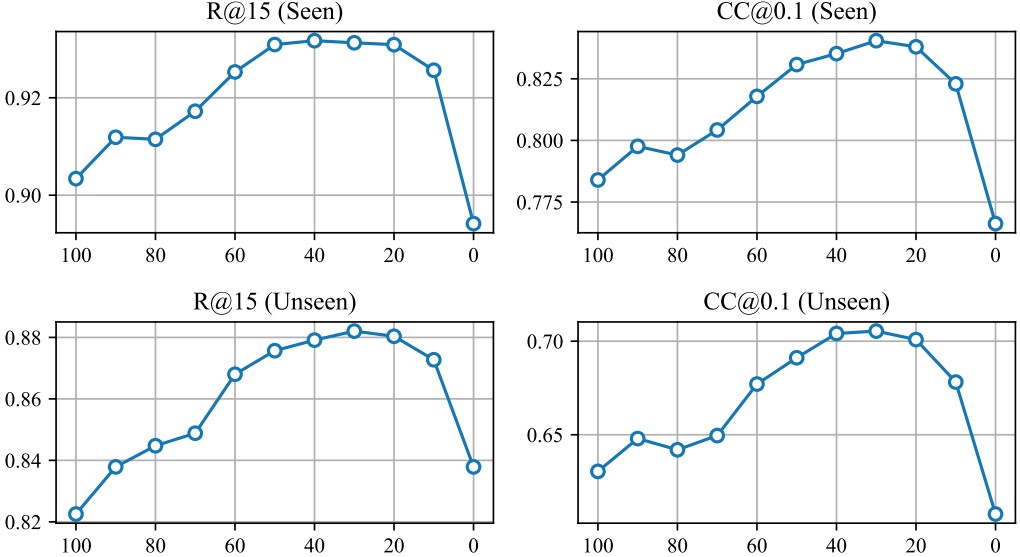

Figure 12: **Early Stopping Ablation for Backward Diffusion on CO3D with 8 Images.** We find empirically that stopping the backward diffusion process early yields slightly improved results. Here, we visualize the accuracy of the predicted $X_0$ after each iteration of backward diffusion, starting from complete noise ($T = 100$) to the final recovered rays $X_0$. For all experiments, we use the $X_0$ predicted at $T = 30$.

| | # of Images | 2 | 3 | 4 | 5 | 6 | 7 | 8 |
|---|---|---|---|---|---|---|---|---|
| | Apple | 87.6 | 83.6 | 83.6 | 83.2 | 84.1 | 83.6 | 84.8 |
| | Backpack | 93.0 | 94.0 | 94.7 | 93.9 | 94.7 | 94.6 | 94.3 |
| | Banana | 90.2 | 90.3 | 91.6 | 91.3 | 91.4 | 91.8 | 92.0 |
| | Baseballbat | 95.7 | 91.0 | 90.2 | 91.1 | 91.9 | 90.3 | 90.2 |
| | Baseballglove | 85.3 | 86.7 | 86.7 | 86.8 | 85.9 | 85.9 | 86.2 |
| | Bench | 94.8 | 94.3 | 93.2 | 94.2 | 94.1 | 94.1 | 93.4 |
| | Bicycle | 92.4 | 92.4 | 92.0 | 93.9 | 93.7 | 93.5 | 94.0 |
| | Bottle | 87.2 | 84.1 | 86.5 | 86.2 | 84.2 | 85.0 | 85.4 |
| | Bowl | 90.5 | 92.2 | 92.4 | 91.4 | 91.9 | 92.5 | 91.9 |
| | Broccoli | 73.8 | 71.9 | 77.2 | 76.2 | 78.1 | 78.1 | 77.9 |
| | Cake | 81.8 | 83.4 | 81.6 | 84.3 | 83.8 | 84.0 | 83.6 |
| | Car | 90.0 | 90.9 | 91.0 | 91.3 | 91.5 | 90.5 | 90.3 |
| | Carrot | 86.1 | 87.6 | 87.4 | 88.1 | 89.1 | 89.0 | 88.7 |
| | Cellphone | 87.6 | 87.7 | 88.3 | 88.1 | 88.9 | 89.6 | 89.3 |
| | Chair | 97.5 | 98.2 | 98.4 | 98.3 | 98.4 | 98.7 | 98.6 |
| | Cup | 80.0 | 77.6 | 77.1 | 78.0 | 78.9 | 77.3 | 77.0 |
| | Donut | 71.2 | 76.5 | 74.7 | 76.1 | 74.2 | 75.8 | 75.4 |
| Seen Categories | Hairdryer | 90.6 | 93.2 | 93.9 | 94.7 | 94.7 | 94.5 | 94.4 |
| | Handbag | 85.1 | 85.1 | 86.8 | 87.5 | 87.6 | 87.9 | 87.1 |
| | Hydrant | 98.0 | 96.9 | 95.9 | 96.7 | 96.8 | 96.7 | 97.5 |
| | Keyboard | 95.4 | 95.4 | 94.2 | 94.7 | 94.5 | 94.7 | 94.7 |
| | Laptop | 95.6 | 96.0 | 96.1 | 96.6 | 96.5 | 96.5 | 96.4 |
| | Microwave | 88.0 | 86.1 | 85.3 | 83.9 | 85.2 | 86.1 | 86.1 |
| | Motorcycle | 92.4 | 92.9 | 93.3 | 94.0 | 94.6 | 94.6 | 94.1 |
| | Mouse | 93.5 | 93.6 | 94.4 | 94.0 | 94.5 | 94.1 | 94.5 |
| | Orange | 75.7 | 75.4 | 73.7 | 73.6 | 74.9 | 75.2 | 74.7 |
| | Parkingmeter | 86.7 | 84.4 | 79.4 | 80.0 | 87.3 | 80.8 | 78.2 |
| | Pizza | 92.4 | 92.1 | 89.8 | 92.8 | 92.8 | 92.8 | 94.2 |
| | Plant | 83.9 | 84.4 | 85.4 | 85.5 | 84.8 | 85.4 | 85.8 |
| | Stopsign | 86.5 | 88.0 | 89.4 | 87.5 | 88.1 | 87.2 | 87.5 |
| | Teddybear | 92.0 | 93.8 | 94.4 | 94.4 | 94.8 | 95.1 | 95.3 |
| | Toaster | 99.2 | 98.4 | 98.5 | 99.2 | 99.0 | 98.8 | 99.0 |
| | Toilet | 97.2 | 97.0 | 95.6 | 96.5 | 97.1 | 96.5 | 96.5 |
| | Toybus | 92.3 | 93.1 | 91.9 | 93.2 | 91.5 | 92.4 | 93.0 |
| | Toyplane | 79.5 | 80.0 | 80.7 | 81.9 | 82.4 | 82.9 | 82.5 |
| | Toytrain | 90.6 | 89.2 | 91.5 | 91.6 | 89.9 | 91.8 | 90.8 |
| | Toytruck | 89.4 | 87.9 | 88.9 | 89.2 | 89.9 | 90.2 | 89.7 |
| | Tv | 100.0 | 100.0 | 98.9 | 97.3 | 99.1 | 98.1 | 98.3 |
| | Umbrella | 88.4 | 90.3 | 89.3 | 90.1 | 90.5 | 90.6 | 89.9 |
| | Vase | 85.2 | 82.7 | 85.0 | 84.4 | 84.6 | 84.7 | 85.1 |
| | Wineglass | 77.5 | 78.1 | 78.7 | 78.0 | 78.6 | 78.7 | 78.1 |
| | Ball | 61.7 | 62.8 | 62.8 | 62.9 | 62.3 | 62.2 | 62.1 |
| | Book | 81.7 | 83.5 | 84.7 | 86.0 | 85.3 | 86.2 | 85.1 |
| | Couch | 86.8 | 88.0 | 85.3 | 87.8 | 86.9 | 87.7 | 88.2 |
| | Frisbee | 75.2 | 75.5 | 76.8 | 78.0 | 77.2 | 77.0 | 77.8 |
| Unseen Categories | Hotdog | 75.7 | 72.9 | 75.7 | 73.3 | 73.2 | 74.6 | 75.3 |
| | Kite | 69.2 | 71.0 | 72.7 | 73.8 | 76.8 | 77.9 | 77.7 |
| | Remote | 88.0 | 90.4 | 92.2 | 93.0 | 91.9 | 93.5 | 92.6 |
| | Sandwich | 79.5 | 79.0 | 78.8 | 78.0 | 78.0 | 78.5 | 78.0 |
| | Skateboard | 77.8 | 77.8 | 80.7 | 85.1 | 85.7 | 85.5 | 85.6 |
| | Suitcase | 94.8 | 95.6 | 96.3 | 96.1 | 95.7 | 95.6 | 95.9 |

Table 12: **Per-category Camera Rotation Accuracy on CO3D (@ 15°) for Ray Regression.**

| # of Images | 2 | 3 | 4 | 5 | 6 | 7 | 8 |
|---|---|---|---|---|---|---|---|
| Apple | 100.0 | 92.9 | 87.3 | 81.4 | 78.5 | 74.7 | 74.1 |
| Backpack | 100.0 | 93.5 | 90.7 | 87.7 | 85.1 | 82.7 | 81.4 |
| Banana | 100.0 | 91.0 | 84.9 | 79.6 | 76.2 | 74.0 | 74.1 |
| Baseballbat | 100.0 | 90.5 | 78.2 | 71.7 | 68.6 | 68.8 | 64.6 |
| Baseballglove | 100.0 | 93.3 | 86.7 | 81.3 | 81.1 | 77.1 | 74.7 |
| Bench | 100.0 | 94.7 | 91.5 | 89.3 | 87.5 | 85.6 | 84.3 |
| Bicycle | 100.0 | 94.7 | 92.5 | 90.6 | 88.2 | 85.7 | 85.2 |
| Bottle | 100.0 | 91.6 | 86.6 | 83.8 | 79.3 | 78.1 | 76.4 |
| Bowl | 100.0 | 91.7 | 89.6 | 86.5 | 85.7 | 84.8 | 84.3 |
| Broccoli | 100.0 | 89.9 | 82.3 | 76.7 | 73.5 | 69.9 | 67.7 |
| Cake | 100.0 | 87.7 | 82.8 | 76.4 | 73.8 | 71.2 | 67.9 |
| Car | 100.0 | 92.1 | 89.9 | 87.9 | 87.6 | 86.4 | 85.3 |
| Carrot | 100.0 | 90.1 | 83.0 | 77.7 | 75.1 | 72.1 | 70.2 |
| Cellphone | 100.0 | 91.2 | 84.2 | 76.1 | 73.7 | 73.1 | 70.5 |
| Chair | 100.0 | 97.5 | 96.9 | 95.6 | 94.9 | 94.1 | 93.5 |
| Cup | 100.0 | 84.8 | 75.9 | 73.0 | 70.2 | 65.5 | 63.4 |
| Donut | 100.0 | 83.7 | 68.2 | 65.9 | 64.7 | 62.7 | 62.6 |
| Hairdryer | 100.0 | 95.6 | 91.3 | 87.4 | 85.6 | 82.7 | 80.8 |
| Handbag | 100.0 | 90.7 | 84.6 | 80.8 | 76.4 | 75.4 | 72.4 |
| Hydrant | 100.0 | 97.9 | 96.0 | 94.7 | 94.1 | 93.8 | 92.2 |
| Keyboard | 100.0 | 91.5 | 83.3 | 79.8 | 77.4 | 74.5 | 74.0 |
| Laptop | 100.0 | 94.9 | 88.0 | 86.7 | 82.7 | 80.2 | 78.8 |
| Microwave | 100.0 | 89.3 | 79.4 | 77.4 | 74.0 | 72.3 | 71.3 |
| Motorcycle | 100.0 | 96.8 | 94.7 | 92.9 | 91.8 | 91.4 | 90.3 |
| Mouse | 100.0 | 95.6 | 88.8 | 85.5 | 82.2 | 79.8 | 75.3 |
| Orange | 100.0 | 85.9 | 73.5 | 68.4 | 63.2 | 60.9 | 59.5 |
| Parkingmeter | 100.0 | 82.2 | 73.3 | 72.7 | 73.9 | 68.1 | 62.5 |
| Pizza | 100.0 | 92.1 | 83.3 | 80.4 | 78.9 | 77.7 | 74.8 |
| Plant | 100.0 | 90.5 | 84.7 | 79.5 | 76.9 | 74.6 | 73.2 |
| Stopsign | 100.0 | 90.1 | 84.9 | 80.8 | 77.5 | 72.2 | 73.5 |
| Teddybear | 100.0 | 97.0 | 92.9 | 90.8 | 88.2 | 86.8 | 85.8 |
| Toaster | 100.0 | 97.3 | 96.9 | 95.6 | 95.5 | 93.1 | 93.2 |
| Toilet | 100.0 | 90.3 | 82.6 | 80.8 | 76.6 | 74.4 | 72.6 |
| Toybus | 100.0 | 96.7 | 88.5 | 88.0 | 85.0 | 83.3 | 82.9 |
| Toyplane | 100.0 | 86.5 | 77.9 | 73.0 | 70.5 | 69.1 | 66.9 |
| Toytrain | 100.0 | 90.4 | 87.3 | 83.1 | 79.3 | 79.3 | 75.7 |
| Toytruck | 100.0 | 91.5 | 85.1 | 82.7 | 81.4 | 80.1 | 77.0 |
| Tv | 100.0 | 95.6 | 91.7 | 84.0 | 83.3 | 84.8 | 87.5 |
| Umbrella | 100.0 | 95.2 | 89.0 | 85.4 | 85.4 | 83.6 | 82.0 |
| Vase | 100.0 | 89.0 | 84.0 | 79.0 | 76.3 | 75.9 | 74.5 |
| Wineglass | 100.0 | 86.0 | 80.0 | 74.4 | 73.5 | 71.8 | 69.3 |
| Ball | 100.0 | 79.7 | 65.0 | 58.8 | 53.1 | 51.0 | 48.5 |
| Book | 100.0 | 90.5 | 83.0 | 79.2 | 75.1 | 75.0 | 69.9 |
| Couch | 100.0 | 79.6 | 66.9 | 64.2 | 60.9 | 58.1 | 57.0 |
| Frisbee | 100.0 | 84.0 | 75.0 | 70.9 | 68.7 | 64.2 | 64.6 |
| Hotdog | 100.0 | 67.6 | 63.2 | 51.1 | 49.5 | 48.8 | 48.0 |
| Kite | 100.0 | 71.8 | 60.4 | 57.4 | 52.8 | 50.0 | 50.6 |
| Remote | 100.0 | 92.1 | 87.7 | 83.0 | 80.5 | 80.3 | 77.8 |
| Sandwich | 100.0 | 89.0 | 81.6 | 76.7 | 73.0 | 71.7 | 71.4 |
| Skateboard | 100.0 | 87.0 | 80.3 | 76.4 | 72.4 | 66.2 | 66.0 |
| Suitcase | 100.0 | 95.3 | 93.3 | 90.5 | 88.2 | 87.3 | 85.5 |

*Seen Categories* (rows Apple–Wineglass), *Unseen Categories* (rows Ball–Suitcase)

Table 13: **Per-category Camera Center Accuracy on CO3D (@ 0.1) for Ray Regression.**

| | # of Images | 2 | 3 | 4 | 5 | 6 | 7 | 8 |
|---|---|---|---|---|---|---|---|---|
| | Apple | 92.8 | 91.3 | 91.3 | 92.2 | 91.8 | 91.5 | 91.6 |
| | Backpack | 94.9 | 95.5 | 95.3 | 95.8 | 96.2 | 95.7 | 96.0 |
| | Banana | 92.2 | 93.2 | 94.4 | 95.1 | 95.8 | 96.0 | 96.5 |
| | Baseballbat | 97.1 | 95.2 | 94.8 | 94.6 | 95.0 | 95.8 | 94.4 |
| | Baseballglove | 89.3 | 92.0 | 90.0 | 90.3 | 90.8 | 91.1 | 90.7 |
| | Bench | 98.0 | 98.4 | 97.5 | 97.4 | 97.5 | 97.8 | 97.3 |
| | Bicycle | 94.4 | 92.5 | 93.5 | 94.4 | 95.1 | 95.3 | 96.3 |
| | Bottle | 91.6 | 90.8 | 90.9 | 91.0 | 91.3 | 92.0 | 92.6 |
| | Bowl | 91.5 | 93.2 | 93.4 | 93.1 | 93.8 | 93.9 | 93.8 |
| | Broccoli | 80.0 | 83.8 | 85.3 | 85.6 | 86.1 | 86.7 | 86.9 |
| | Cake | 91.6 | 91.1 | 91.3 | 91.4 | 90.9 | 91.3 | 90.9 |
| | Car | 92.6 | 93.3 | 92.3 | 93.2 | 93.5 | 93.1 | 93.3 |
| | Carrot | 88.7 | 90.4 | 92.0 | 92.5 | 92.6 | 92.5 | 92.2 |
| | Cellphone | 92.0 | 92.9 | 91.9 | 92.3 | 92.4 | 93.0 | 92.6 |
| | Chair | 98.9 | 98.9 | 98.8 | 99.3 | 99.1 | 99.4 | 99.4 |
| | Cup | 84.4 | 82.5 | 85.1 | 83.8 | 84.1 | 84.5 | 84.4 |
| | Donut | 89.6 | 86.9 | 88.5 | 87.8 | 88.9 | 87.9 | 88.6 |
| | Hairdryer | 93.5 | 95.4 | 96.5 | 95.9 | 96.4 | 97.0 | 96.9 |
| Seen Categories | Handbag | 88.3 | 89.6 | 91.0 | 90.9 | 91.4 | 91.8 | 91.2 |
| | Hydrant | 97.6 | 98.0 | 97.7 | 98.1 | 99.0 | 98.8 | 99.1 |
| | Keyboard | 94.4 | 95.3 | 95.3 | 95.6 | 95.4 | 96.0 | 96.2 |
| | Laptop | 97.1 | 97.0 | 96.5 | 96.8 | 97.1 | 97.2 | 97.1 |
| | Microwave | 89.6 | 88.3 | 88.0 | 87.6 | 87.6 | 88.5 | 88.7 |
| | Motorcycle | 95.6 | 97.2 | 96.7 | 96.7 | 96.9 | 97.1 | 96.6 |
| | Mouse | 97.1 | 96.6 | 96.9 | 97.8 | 97.3 | 97.5 | 97.8 |
| | Orange | 85.4 | 86.9 | 84.1 | 86.4 | 85.9 | 85.5 | 85.7 |
| | Parkingmeter | 76.7 | 84.4 | 90.0 | 90.0 | 91.8 | 93.5 | 92.4 |
| | Pizza | 95.2 | 95.9 | 93.7 | 95.1 | 95.3 | 95.0 | 94.7 |
| | Plant | 91.2 | 91.6 | 92.8 | 93.8 | 93.4 | 93.6 | 94.1 |
| | Stopsign | 90.2 | 89.8 | 89.9 | 89.5 | 90.5 | 89.4 | 89.8 |
| | Teddybear | 94.1 | 96.6 | 96.9 | 97.3 | 97.7 | 97.6 | 98.0 |
| | Toaster | 97.6 | 98.4 | 98.5 | 99.2 | 99.0 | 98.8 | 99.5 |
| | Toilet | 99.3 | 97.7 | 97.1 | 97.2 | 97.0 | 96.8 | 96.9 |
| | Toybus | 89.2 | 91.8 | 92.1 | 94.4 | 91.7 | 92.7 | 92.9 |
| | Toyplane | 84.1 | 85.3 | 85.5 | 86.3 | 86.5 | 86.7 | 85.8 |
| | Toytrain | 93.1 | 93.3 | 92.2 | 93.3 | 92.4 | 93.4 | 93.1 |
| | Toytruck | 88.1 | 90.1 | 88.6 | 89.1 | 89.9 | 91.0 | 91.1 |
| | Tv | 100.0 | 100.0 | 100.0 | 98.0 | 100.0 | 100.0 | 100.0 |
| | Umbrella | 90.8 | 92.5 | 93.1 | 92.6 | 92.4 | 93.3 | 93.7 |
| | Vase | 87.1 | 90.4 | 90.8 | 90.8 | 90.8 | 90.5 | 91.3 |
| | Wineglass | 87.0 | 85.7 | 85.6 | 86.8 | 87.4 | 86.8 | 86.6 |
| | Ball | 73.6 | 74.0 | 74.6 | 74.8 | 76.0 | 74.0 | 75.6 |
| | Book | 90.4 | 90.6 | 91.6 | 91.9 | 92.6 | 92.5 | 92.7 |
| | Couch | 90.8 | 89.7 | 89.8 | 92.1 | 90.4 | 90.9 | 90.3 |
| | Frisbee | 75.2 | 78.1 | 79.1 | 83.0 | 82.3 | 82.7 | 84.2 |
| | Hotdog | 70.0 | 80.0 | 80.2 | 78.4 | 78.9 | 79.9 | 81.3 |
| Unseen Categories | Kite | 76.9 | 77.7 | 78.7 | 79.8 | 81.1 | 82.5 | 82.7 |
| | Remote | 92.0 | 94.0 | 95.9 | 94.8 | 95.4 | 95.1 | 95.5 |
| | Sandwich | 87.0 | 87.3 | 87.0 | 88.1 | 88.6 | 89.3 | 90.3 |
| | Skateboard | 83.3 | 86.7 | 88.9 | 88.8 | 89.5 | 90.5 | 90.1 |
| | Suitcase | 95.6 | 97.9 | 97.5 | 97.6 | 97.7 | 98.0 | 97.9 |

Table 14: **Per-category Camera Rotation Accuracy on CO3D (@ 15°) for Ray Diffusion.**

|  | # of Images | 2 | 3 | 4 | 5 | 6 | 7 | 8 |
|---|---|---|---|---|---|---|---|---|
| Seen Categories | Apple | 100.0 | 96.1 | 92.4 | 89.5 | 87.1 | 85.7 | 84.5 |
|  | Backpack | 100.0 | 96.1 | 93.8 | 91.6 | 90.0 | 87.1 | 86.9 |
|  | Banana | 100.0 | 94.1 | 88.9 | 84.7 | 82.5 | 81.5 | 81.1 |
|  | Baseballbat | 100.0 | 92.4 | 81.8 | 78.0 | 77.4 | 77.6 | 71.4 |
|  | Baseballglove | 100.0 | 92.0 | 85.3 | 83.7 | 81.8 | 79.6 | 79.0 |
|  | Bench | 100.0 | 97.3 | 96.1 | 93.7 | 92.5 | 93.6 | 93.8 |
|  | Bicycle | 100.0 | 94.9 | 94.5 | 92.8 | 90.3 | 89.2 | 89.7 |
|  | Bottle | 100.0 | 94.5 | 93.1 | 90.1 | 89.4 | 89.9 | 88.2 |
|  | Bowl | 100.0 | 92.7 | 90.6 | 89.3 | 89.3 | 88.3 | 87.5 |
|  | Broccoli | 100.0 | 95.0 | 88.8 | 84.3 | 81.5 | 81.1 | 78.9 |
|  | Cake | 100.0 | 93.9 | 90.6 | 87.7 | 83.5 | 83.1 | 80.6 |
|  | Car | 100.0 | 94.9 | 92.5 | 91.1 | 91.1 | 90.5 | 90.4 |
|  | Carrot | 100.0 | 93.4 | 88.0 | 84.5 | 82.2 | 79.6 | 79.5 |
|  | Cellphone | 100.0 | 94.4 | 87.8 | 83.6 | 80.5 | 77.8 | 75.9 |
|  | Chair | 100.0 | 98.1 | 97.2 | 96.8 | 96.5 | 95.9 | 95.7 |
|  | Cup | 100.0 | 88.9 | 83.5 | 81.5 | 80.1 | 77.7 | 77.9 |
|  | Donut | 100.0 | 90.4 | 87.4 | 85.6 | 85.2 | 81.7 | 80.7 |
|  | Hairdryer | 100.0 | 97.4 | 95.6 | 93.1 | 91.2 | 90.4 | 89.2 |
|  | Handbag | 100.0 | 93.9 | 91.2 | 86.9 | 85.2 | 83.9 | 81.9 |
|  | Hydrant | 100.0 | 98.7 | 98.0 | 97.1 | 97.7 | 97.3 | 96.7 |
|  | Keyboard | 100.0 | 94.4 | 86.1 | 83.6 | 81.5 | 80.3 | 80.1 |
|  | Laptop | 100.0 | 95.3 | 88.4 | 87.0 | 84.8 | 83.2 | 82.3 |
|  | Microwave | 100.0 | 90.7 | 82.8 | 79.4 | 76.1 | 76.2 | 75.4 |
|  | Motorcycle | 100.0 | 98.3 | 97.3 | 95.0 | 94.9 | 94.8 | 94.4 |
|  | Mouse | 100.0 | 98.2 | 94.9 | 91.5 | 89.7 | 88.5 | 85.9 |
|  | Orange | 100.0 | 92.3 | 84.9 | 81.8 | 77.5 | 75.3 | 74.4 |
|  | Parkingmeter | 100.0 | 87.8 | 93.3 | 84.7 | 88.9 | 86.2 | 85.0 |
|  | Pizza | 100.0 | 92.4 | 90.7 | 87.4 | 84.8 | 83.3 | 81.3 |
|  | Plant | 100.0 | 95.7 | 92.3 | 89.7 | 88.0 | 86.6 | 86.3 |
|  | Stopsign | 100.0 | 93.6 | 87.2 | 83.3 | 80.7 | 77.1 | 77.5 |
|  | Teddybear | 100.0 | 97.9 | 95.5 | 94.3 | 92.8 | 91.9 | 91.5 |
|  | Toaster | 100.0 | 98.5 | 98.0 | 98.7 | 97.9 | 96.2 | 97.4 |
|  | Toilet | 100.0 | 91.7 | 87.9 | 83.3 | 80.8 | 79.2 | 77.5 |
|  | Toybus | 100.0 | 95.1 | 88.1 | 89.5 | 83.6 | 84.5 | 85.9 |
|  | Toyplane | 100.0 | 87.0 | 80.8 | 76.8 | 74.4 | 73.5 | 72.2 |
|  | Toytrain | 100.0 | 92.3 | 87.0 | 85.9 | 82.5 | 83.8 | 79.7 |
|  | Toytruck | 100.0 | 90.4 | 85.6 | 80.8 | 80.5 | 80.1 | 78.1 |
|  | Tv | 100.0 | 100.0 | 100.0 | 98.7 | 97.8 | 95.2 | 96.7 |
|  | Umbrella | 100.0 | 95.3 | 92.9 | 90.5 | 90.5 | 88.7 | 88.5 |
|  | Vase | 100.0 | 95.0 | 91.5 | 88.2 | 87.1 | 85.9 | 85.0 |
|  | Wineglass | 100.0 | 90.6 | 88.1 | 84.6 | 83.2 | 82.2 | 81.4 |
| Unseen Categories | Ball | 100.0 | 85.9 | 75.1 | 70.4 | 67.1 | 62.7 | 63.5 |
|  | Book | 100.0 | 93.7 | 88.0 | 85.3 | 84.2 | 82.5 | 82.3 |
|  | Couch | 100.0 | 83.3 | 74.9 | 71.1 | 65.7 | 64.5 | 61.5 |
|  | Frisbee | 100.0 | 84.3 | 80.2 | 75.8 | 72.5 | 70.7 | 70.5 |
|  | Hotdog | 100.0 | 77.6 | 67.5 | 60.0 | 59.3 | 59.2 | 55.5 |
|  | Kite | 100.0 | 77.2 | 63.8 | 58.8 | 54.2 | 52.1 | 53.5 |
|  | Remote | 100.0 | 94.0 | 91.9 | 87.4 | 85.8 | 84.3 | 83.4 |
|  | Sandwich | 100.0 | 95.0 | 91.5 | 88.9 | 86.4 | 84.5 | 84.9 |
|  | Skateboard | 100.0 | 88.1 | 84.2 | 79.3 | 73.7 | 72.7 | 67.9 |
|  | Suitcase | 100.0 | 97.5 | 94.3 | 93.0 | 91.8 | 90.9 | 90.8 |

Table 15: **Per-category Camera Center Accuracy on CO3D (@ 0.1) for Ray Diffusion.**

