# OpenReview forum: "Cameras as Rays: Pose Estimation via Ray Diffusion"
_ICLR.cc/2024/Conference — ICLR 2024 oral_

### Official Review · Reviewer_mia4 · 2023-10-20

**Soundness:** 2 fair
**Presentation:** 3 good
**Contribution:** 3 good
**Rating:** 5
**Confidence:** 5

**Summary:**

This paper introduces a new method for sparse-view camera pose estimation. Instead of parameterizing a camera model as an intrinsic matrix and an extrinsic matrix, the authors propose to over-parameterize the camera as a collection of rays. The intrinsic and extrinsic are computed by solving a least-squares problem. A diffusion network is presented to predict the ray parameters. The method achieves state-of-the-art performance on the Co3D dataset.

**Strengths:**

•	The idea of over parameterization is novel. It enables robust camera pose estimation by involving least-squares optimization. Ideally, the method has the potential to predict the camera pose from a single RGB image since the ray representation does not rely on multi-view information.

•	The paper is well-written and easy to follow. The experimental results show much better camera pose estimation performance compared with some existing approaches.

**Weaknesses:**

As reported in Table 1, it seems that the presented over parameterization method plays a crucial role in the framework. The performance of Ray Regression (Ours) surpasses that of R+T Regression by a considerable margin. The diffusion model only results in a 3.8% improvement in the case of two images. To my understanding, the superior performance is primarily attributed to the least-squares optimization which accounts for a robust estimation. However, it is still quite confusing why the pose estimation benefits from the ray representation.

Basically, the idea is to regress a ray represented as a 6D vector for each patch in the RGB image. It is arguably more challenging than predicting R and T. The difficulty lies in two aspects. First, it is a dense prediction problem. Second, it regresses 3D information from RGB images. One could also predict the corresponding 2D coordinates in the right image for each patch in the left image as an alternative. Intuitively, it is easier to predict 2D coordinates than 6D ray vectors. The authors argue that such a method could struggle in sparse view settings due to insufficient image overlap to find correspondences. It is unclear why the presented method is able to achieve better robustness.

Moreover, it is confusing why the presented method can recover the translation. According to Eq.3, m is coupled with the translation t. Predicting m then demands a requirement of capturing information about the camera translation. However, the actual input of the network is a cropped image. The information regarding t loses after the cropping.

The authors only conducted experiments on the Co3D dataset, which makes the evaluation not convincing enough. There are several datasets that have been widely used in the literature such as Megadepth, ScanNet, and HPatches. It would be beneficial if the authors could show the effectiveness of the over parameterization on such datasets.

According to Eq.7, the patches of all available images are jointly processed, which is computationally expensive. As reported by the authors, training the diffusion model takes four days on 8 A6000 GPUs, which is much slower than RelPose and RelPose++.

**Questions:**

•	Most of the equations in this paper make sense to me, but the explanation of Eq.5 is a bit confusing. What is the “identity” camera? Are there any constraints on this equation? Does it still hold when the image depicts multiple planes?

•	As shown in Table 1, sometimes, the performance of the presented method decreases when more images are involved. By contrast, the performance of most competitors such as RelPose++ consistently becomes better with more images. I was wondering why the method is not compatible with multi-view images.

---

> ### Author Response · Authors · 2023-11-16
> **Response Part 1**
>
> > As reported in Table 1, it seems that the presented over parameterization method plays a crucial role in the framework. The performance of Ray Regression (Ours) surpasses that of R+T Regression by a considerable margin. The diffusion model only results in a 3.8% improvement in the case of two images. To my understanding, the superior performance is primarily attributed to the least-squares optimization which accounts for a robust estimation. However, it is still quite confusing why the pose estimation benefits from the ray representation.
>
> We agree with the reviewer’s assessment that over-parameterization plays a crucial role, but the other important source of improvement is the tight coupling between the ray estimation and local features. We respectfully disagree with the reviewer’s statement that performance is “primarily attributed to the least-squares optimization which accounts for a robust estimation.” To verify this, we conduct an experiment that still makes use of our patch-predicted rays but reduces the “robustness” of the least squares optimization. Specifically, we use a randomly sampled subset of the 256 predicted rays per camera to recover the camera extrinsics. We compute the rotation accuracy for our Ray Regression model on unseen object categories for N=8 images:
>
>
> | # of Rays                        | 6    | 16   | 26   | 56   | 106    | 256  |
> |----------------------------------|------|------|------|------|------|------|
> | Ray Regression Rotation Accuracy | 73.8 | 81.2 | 81.2 | 81.4 | 81.8 | 81.8 |
> | Ray Regression Camera Center Accuracy | 60.7 | 61.8 |  61.9 |  61.4 | 61.8 | 61.6 |
>
> As we can see, accuracy goes up very quickly, even with rather few rays. Of course, the least-squares optimization gives robustness (e.g. comparing 6 rays vs 16 rays), but we believe the primary gain is from the accurately estimated rays by the local-level joint reasoning of our network.
>
> > Basically, the idea is to regress a ray represented as a 6D vector for each patch in the RGB image. It is arguably more challenging than predicting R and T. The difficulty lies in two aspects. First, it is a dense prediction problem. Second, it regresses 3D information from RGB images. One could also predict the corresponding 2D coordinates in the right image for each patch in the left image as an alternative. Intuitively, it is easier to predict 2D coordinates than 6D ray vectors. The authors argue that such a method could struggle in sparse view settings due to insufficient image overlap to find correspondences. It is unclear why the presented method is able to achieve better robustness.
>
> We agree that predicting dense 2D correspondences between image pairs is an interesting alternative. However, converting such 2D correspondences to 3D cameras (in the presence of uncertainty, partial visibility, etc) is still challenging. Moreover, this is fundamentally a pairwise prediction task and is hard to extend to multiple images where the question of which images should be matched is also uncertain. Because our solution is not restricted to such pairwise reasoning, it bypasses this issue altogether. In addition, the representation inferred is directly convertible to global cameras.
>
> We also note that our COLMAP baseline essentially does the sparser version of such a 2D correspondence-to-camera pipeline using a state-of-the-art feature matcher. Such methods lack robustness because of insufficient 2D correspondences.
>
> > Moreover, it is confusing why the presented method can recover the translation. According to Eq.3, m is coupled with the translation t. Predicting m then demands a requirement of capturing information about the camera translation. However, the actual input of the network is a cropped image. The information regarding t loses after the cropping.
>
> Thank you for pointing this out. We do provide the bounding box information of the crop to the network which is important for reasoning about translation (as the reviewer notes). Specifically, we concatenate the pixel coordinate of each patch (in normalized device coordinates with respect to the uncropped image) to the spatial features for both the ray regression and ray diffusion models that we train.
>
> We apologize for missing this detail in the earlier version and have now clarified this in the text and added the pixel coordinates to Eqs 7 and 11.

---

> ### Author Response · Authors · 2023-11-16
> **Response Part 2**
>
> > According to Eq.7, the patches of all available images are jointly processed, which is computationally expensive. As reported by the authors, training the diffusion model takes four days on 8 A6000 GPUs, which is much slower than RelPose and RelPose++
>
> The per-iteration training speed of our method is similar to that of RelPose++, but we train for half the iterations, so our total training time is about half that of RelPose++ (4 days vs 8 days of training on 8 A6000s). As we report in Table 6, the inference speed of our Ray Diffusion method is 2.6X faster than RelPose while our Ray Regression method (which requires a single forward pass) is 220X faster.
>
> > the explanation of Eq.5 is a bit confusing. What is the “identity” camera? Are there any constraints on this equation? Does it still hold when the image depicts multiple planes?
>
> By identity camera, we refer to a camera for which both the intrinsics and rotation are identity matrices: $K=I, R=I$. We have now clarified this in the text.
>
> If we have a camera with rotation $R$ and intrinsics $K$, the direction of the ray corresponding to a (homogenous) pixel coordinate $u_i \in \mathbf{P}^2$ can be computed as $R^T K^{-1} u_i$. Given predicted directions $d_i$, we thus need to compute optimal $R, K$ such that $R^T K^{-1} u_i = d_i$,  or equivalently, $(K R) d_i = P d_i = u_i$. Note that this is an ‘equality’ in homogenous representations (up-to-scale). Solving this corresponds to finding the optimal homography matrix such that $ H d_i = u_i$.
>
> Our terminology stems from an alternate interpretation where ‘directions’ can be thought of as points on the plane at infinity (in projective 3D space) i.e. a direction $d_i  \in \mathbf{P}^2$ implies a point $(d_i,0) \in \mathbf{P}^3$. In this interpretation, finding $K,R$ is equivalent to finding the homography that relates  the images of this plane at infinity under the two cameras: identity camera $(K=I, R=I)$ and the regular camera $(K=K, R=R)$ (see Sec 8.5 in Hartley & Zisserman).
>
> Please note that this does not have anything to do with planar surfaces present in the scene; and eq 5 holds for any scene geometry.
>
>
> > As shown in Table 1, sometimes, the performance of the presented method decreases when more images are involved.
>
> This issue appears to be due to a bug when computing our camera intrinsics at training time (see top-level comment). Now that the bug has been fixed, the performance does not decrease with more images as expected, as shown in Table 1. We apologize for the confusion.

---

> ### Comment · Reviewer_mia4 · 2023-11-20
> **Response to Authors**
>
> I sincerely thank the authors for the rebuttal. Some of my concerns have been addressed, but my major concern still exists. The advantages compared to correspondence-based methods are still unclear to me. The authors argued that these methods need pairwise images as input while the presented method predicts the camera pose from the single-view image. However, to estimate the camera pose, the most straightforward idea is to somehow predict the pose from a single image. Since such a single-view prediction doesn't work well, pairwise images are utilized to make the problem easier. In the presented method, the predicted $\mathbf{d}$ represents a direction in the canonical coordinate system, which means the network has to learn the camera rotation from the current view to the canonical view. Moreover, as I mentioned before, the authors propose to learn the direction for each image patch. Intuitively, such a dense prediction from a single view is more challenging. I don't understand why the method achieves promising results while the correspondence-based approach fails.
>
> As updated by the authors, ray diffusion works worse than ray regression in camera rotation estimation, which weakens the claimed contribution. More importantly, the authors have found that there is a strong bias in the CO3D dataset and even predicting a constant rotation works better than RelPose++. This makes the conducted experiments less convincing. Given that the presented method predicts directions in the canonical frame, there is a potential risk of it being overfitted to this observed bias. The authors added some qualitative results on MegaDepth, which are not supportive enough.

---

> > ### Author Response · Authors · 2023-11-20
> > **Clarifications**
> >
> > We thank the reviewer for the response and opportunity for discussion. We believe both concerns raised by the reviewer may stem from possible misunderstandings, and we attempt to clarify those below.
> >
> > - - -
> >
> > ## Clarification on Setup
> >
> > The reviewer stated that *"the presented method predicts the camera pose from the single-view image"*. This would suggest that our method is taking in one image at a time and predicting the corresponding ray bundle: $ \mathcal{R}_i = f(I_i)$.
> >
> > Rather, we wish to clarify that our method predicts the rays from all images $\\{I_1, \ldots, I_N\\}$ **jointly** (See Eqs 7 and 11):
> > $$ \\{\mathcal{R}_1, \ldots, \mathcal{R}_N\\} = f(\\{I_1, \ldots, I_n\\}).$$
> >
> > We do predict one ray per patch, but to reiterate, we predict these in the context of all patches across all images. Thus, the network has the capacity to reason about correspondences implicitly.
> >
> > ## Direct Prediction vs Cameras from Pairwise Correspondences
> >
> > We believe that the reviewer’s concerns are regarding comparing two philosophically different approaches for camera prediction – ‘direct’ vs ‘correspondence based’, and why one may be better than the other.
> >
> > By a ‘direct’ approach, we imply a system that, like ours, predicts a set of cameras given a set of images without constructing pairwise correspondences:
> >  $$ f(I_1, \ldots, I_n) = \\{\Pi_1, \ldots,\Pi_N\\}.$$
> >
> > In contrast, correspondence-based methods (e.g. COLMAP) would infer pairwise correspondences and then recover cameras.
> > $$ f(I_i, I_j) = C_{i\rightarrow j}, \\{C_{i \rightarrow j} \\} \Rightarrow \\{\Pi_1, \ldots,\Pi_N\\} $$
> >
> > We would like to emphasize that ours is not the first work to highlight that learning direct prediction can perform better than correspondence-based methods. In particular, SparsePose and PoseDiffusion both show that their methods (where cameras are defined in a first-image-aligned frame similar to ours) outperform COLMAP. Our approach further improves over these via a different camera parameterization.
> >
> > While we agree that this patchwise ray prediction is a challenging task, unlike the two-stage correspondence approach, it does not require explicit pairwise matching, which is difficult under wide baseline images or textureless objects. Moreover, the correspondence-based methods require a second stage to infer cameras and this can itself be a challenging task.
> >
> > Finally, we would also like to reiterate that in the context of prior prediction methods which outperform correspond-based baselines, our work simply improves the camera representation predicted. The fact that correspondence-based methods perform worse in comparison is an already established fact in these prior methods, and respectfully, should not be an argument against our work in particular.
> >
> >
> > ## Clarification on Additional Results
> >
> > We believe there may be a misunderstanding regarding the new experiments added in our top-level comment. These are **not** updated results on Co3D (which are updated in the pdf in Tables 1 and 2). Instead, the table in our comment corresponds to zero-shot generalization to a scene-level dataset RealEstate10k, which is an experiment used by PoseDiffusion to test such generalization. In the context of this clarification, we address the comments from the reviewer below:
> >
> > > As updated by the authors, ray diffusion works worse than ray regression in camera rotation estimation
> >
> > This is **not** true in general. In our main results in CO3D, RayDiffusion always outperforms RayRegression (accuracy of 88.1% compared to 81.9% for rotation, N=8 for unseen categories). In RealEstate10k, the two perform almost similarly (with Regression marginally better), and we conjecture this is because there is less uncertainty in the data (no symmetry, smaller motions on average).
> >
> >  > More importantly, the authors have found that there is a strong bias in the CO3D dataset and even predicting a constant rotation works better than RelPose++. This makes the conducted experiments less convincing.
> >
> > The bias we identified is in RealEstate10k data, and we only wanted to clarify that this generalization experiment recommended by PoseDiffusion should be taken with a small grain of salt. There is **no** such forward-facing bias in CO3D, which consists of turntable-style object captures, and the conclusions from our main results still stand.
> >
> > > Given that the presented method predicts directions in the canonical frame, there is a potential risk of it being overfitted to this observed bias.
> >
> > Our approach performs better compared to the baselines on this scene-level dataset with different camera distributions than the training dataset CO3D. This in fact shows that it is more robust to such observation biases!
> >
> > - - - -
> >
> > We again wish to thank the reviewer for this opportunity to engage, and would really appreciate if they could indicate whether the above responses helped clarify any misunderstandings. We would be happy to address any concerns that remain.

---

> > > ### Comment · Reviewer_mia4 · 2023-11-22
> > >
> > > I thank the authors for the clarification and I am sorry for the previous misunderstanding. I acknowledge the novelty of this paper and I believe it would be a strong submission if more quantitative results on other datasets such as Megadepth and ScanNet could be reported. I understand it might be impractical to run these experiments in the rebuttal phase, but I have to keep my original rating because the experimental results on a single dataset co3d are not convincing enough to me.

---

> > > > ### Author Response · Authors · 2023-11-23
> > > >
> > > > We thank the reviewer for their response, and are glad to hear the misunderstanding was addressed and that the reviewer does appreciate the technical contributions of our work. However, as the reviewer still indicated they may lean towards rejection, we wish to state our case again before the discussion period ends. Based on the reviewer comments above, the primary concern that now affects their rating is that they believe it is critical to show empirical results on datasets like MegaDepth. While we appreciate the comments that we (and the field) should examine these setups, we feel that setting this new evaluation setup as a barrier for acceptance is an unreasonable (and counterproductive) bar for an emerging research area.
> > > >
> > > > Specifically, learning-based methods for pose prediction with sparse-view/wide-baseline input represent a growing body of work, where prior methods (e.g. RelPose [ECCV 22], SparsePose [CVPR 23], PoseDiffusion [ICCV 23], RelPose++ [3DV 24]) have evaluated object-centric pose estimation in datasets like CO3D. In this commonly adopted setup, which is of practical significance in applications like reconstruction in online marketplaces, our approach clearly outperforms all these baselines. Moreover, following PoseDiffusion, we also demonstrate generalization to a scene-centric dataset RealEstate10k, where our approach again improves over prior work. Given these empirical results in the standard evaluation setup adopted by prior work, combined with the technical contributions (that the reviewer agrees on), we believe that our work clearly represents a step forward for this area of learning-based pose prediction and that the community would benefit from it.
> > > >
> > > > Again, we understand the reviewer may wish to see methods such as ours being evaluated in other scenarios that are commonly studied in the SfM community. But such change occurs gradually, and should not be enforced overnight! For example, PoseDiffusion proposed a scene-centric generalization, and our work now incorporates the preliminary results on MegaDepth (which will hopefully encourage others to follow suit). We would therefore urge the reviewer to reconsider, and respectfully, not enforce a threshold that deviates from common practice in prior works as a criterion against acceptance of our work.

---

> ### Comment · Reviewer_mia4 · 2023-11-23
>
> I disagree with the authors that the comments on the experiments are unreasonable and counterproductive!
>
> First, I don't think the learning-based camera pose estimation is an "emerging" research area. Even for the object-centric scenarios, some studies, such as [A, B, C], conducted experiments on other datasets. The authors argued that their work clearly represents a step forward for the area, but this is overclaimed to me with the experiments conducted on a single dataset.
>
> Second, the authors have already reported some qualitative results on MegaDepth, which means they managed to test the method on MegaDepth's benchmark. The quantitative results are still missing, so I have a reason to wonder if the method can really work on the scene reconstruction dataset. Some studies like [D] reported experimental results on several datasets, using pairwise images as input. The authors asserted the advantages of their method compared to these competitors, but none of those datasets are examined in this paper.
>
> I agree with the authors that these additional experiments cannot be done overnight, so I keep my previous score based on the quality of the current submission. Again, I don't see a point that the rating is unreasonable.
>
> [A] Zhou, Xingyi, et al. "Starmap for category-agnostic keypoint and viewpoint estimation." Proceedings of the European Conference on Computer Vision (ECCV). 2018.
>
> [B] Xiao, Yang, et al. "Pose from shape: Deep pose estimation for arbitrary 3d objects." arXiv preprint arXiv:1906.05105 (2019).
>
> [C] Ahmadyan, Adel, et al. "Objectron: A large scale dataset of object-centric videos in the wild with pose annotations." Proceedings of the IEEE/CVF conference on computer vision and pattern recognition. 2021.
>
> [D] Sun, Jiaming, et al. "LoFTR: Detector-free local feature matching with transformers." Proceedings of the IEEE/CVF conference on computer vision and pattern recognition. 2021.

---

### Official Review · Reviewer_wRVJ · 2023-10-30

**Soundness:** 3 good
**Presentation:** 2 fair
**Contribution:** 2 fair
**Rating:** 5
**Confidence:** 3

**Summary:**

The paper proposes a distributed representation of camera pose which treats a camera as a bundle rays allowing for a tight coupling with spatial image features, which is naturally suited for set-level level transformers. Furthermore, the authors propose a regress-based approach to map image patches to associated rays. To further capture the inherent uncertainties in pose inference, the authors also develop a denoising diffusion model. The experiment on CO3D dataset demonstrate the performance of the proposed method.

**Strengths:**

1.	The authors propose a novel representation of pose that allows a bundle rays to denote camera in the field of sparse-view pose estimation.
2.	To inference the rays, the authors develop a deterministic regression network and a probabilistic diffusion model, and the experiment on the CO3D demonstrates the superior performance.

**Weaknesses:**

1.	The authors announce that the traditional representation of pose maybe suboptimal in neural learning in the part of introduction. However, no further discussion is given. More specific explanation is necessary, and the comparison with the proposed novel representation of pose is also required.
2.	The punctuation is necessary at the end of each equation, please check it carefully.
3.	The authors fail to state more details of the proposed network architecture. Moreover the training detail is also required.
4.	To demonstrate the performance of the proposed novel representation, can authors undertake more experiments on more datasets?
5.	Please check the format of REFERENCES.

**Questions:**

See the weakness part

---

> ### Author Response · Authors · 2023-11-16
>
> >The authors announce that the traditional representation of pose maybe suboptimal in neural learning in the part of introduction. However, no further discussion is given. More specific explanation is necessary, and the comparison with the proposed novel representation of pose is also required.
>
> We apologize if this was unclear, but by “traditional representation of pose” in neural learning, we meant global parameterizations of 6D pose in the form of rotation and translation. Our experiments do highlight the benefits of our approach which represents cameras as rays compared to representative approaches which represent cameras using the “traditional representation.” In particular, in Table 1, we demonstrate that our representation improves rotation accuracy by 58% for regression-based approaches (Ray Regression vs R+T Regression) and 22% for diffusion-based approaches (Ray Diffusion vs PoseDiffusion w/o GGS). The improvement of our representation over comparable methods that use the traditional global representation of cameras holds for camera center accuracies and all numbers of images.
>
> We hypothesize that this improvement is due to the following factors:
> 1. Predicting bundles of rays is particularly well-suited to transformer-based set-to-set inference.
> 2. Local features enable reasoning about low-level features like correspondences that are not possible in existing global-feature parameterizations in prior work.
>
> > The authors fail to state more details of the proposed network architecture. Moreover the training detail is also required.
>
> We use standard network architectures: DINOv2 for feature extraction and DiT for regressing and diffusing camera rays. We have included additional training details such as dataset preparation, model architecture, and diffusion hyperparameters in Section 3.4 Implementation Details. We would be happy to clarify any other details, and we will be releasing code to ensure reproducibility.
>
> > To demonstrate the performance of the proposed novel representation, can authors undertake more experiments on more datasets?
>
> We have added experiments on zero-shot generalization to RealEstate10K (following PoseDiffusion) as well as newly trained results on MegaDepth (see top-level comment). We find that our method has significantly better zero-shot generalization compared to previous methods in this scene-centric setup.
>
> > The punctuation is necessary at the end of each equation, please check it carefully. Please check the format of REFERENCES.
>
> Thank you for these suggestions. We have added punctuation to the end of equations and updated the reference format.

---

> > ### Comment · Reviewer_wRVJ · 2023-11-20
> >
> > Great, the authors have addressed my concerns

---

> > > ### Author Response · Authors · 2023-11-20
> > > **Thank you!**
> > >
> > > Thank you for your response. We are glad to hear that your concerns were addressed. We would really appreciate it if you could revise your ratings in light of this.

---

### Official Review · Reviewer_v24e · 2023-10-31

**Soundness:** 3 good
**Presentation:** 3 good
**Contribution:** 4 excellent
**Rating:** 8
**Confidence:** 4

**Summary:**

This paper proposes a novel method for estimating wide baseline camera poses from mutliview imagery by representing cameras as a bundle of rays through image pixels.  The rays are directly regressed from local image patches that they pass through using a vision transformer and then made more consistent with neighboring rays using diffusion applied to an image of the rays.  The bundle of rays can be converted to a standard pinhole camera model by a DLT fit of the camera parameters to the rays.  The authors train regression and diffusion models on data from the CO3Dv2 dataset and evaluate the models on held out data from that same dataset.  The method is compared to several other recent methods for camera pose estimation on the same data and demonstrates improvements in rotation and pose accuracy metrics compared to the prior work.  The primary contribution of the work is showing that regression of rays can result in more accurate camera models than trying to directly regress camera parameters as done in prior work.

**Strengths:**

The strengths of this paper are the novelty of the approach and the quality of results, which together are likely to have a significant impact in the field of wide baseline camera estimation.  Directly regressing rays intuitively makes sense as they more suited to regression by a neural network, since each ray depends on more local image information.  The paper makes this point clear and backs it up with experimental results.  Overall, the paper is written clearly and is easy to understand.

**Weaknesses:**

The main weakness of this paper is the somewhat contrived and limited dataset and metrics used in the experimental results.  The CO3D dataset consists of many turntable-like videos with a camera orbiting in a circle around a single object of interest at an approximately fixed distance.  The variability of camera poses is quite limited compared to images in the wild.  Furthermore the image is tightly cropped around the object of interest.  This tight cropping ensures that most rays sampled pass through this common object in all views, which could provide added benefit to the propose approach.  The cropping also provides a disadvantage to feature matching approaches such as used by COLMAP.  COLMAP benefits from having a larger context of the scene with more features to match.  However, the authors are just duplicating the experimental setup from prior work (RelPose), so they are not entirely at fault for these decisions.

The proposed algorithm also, presumably, does not estimate precise camera parameters and would need a further bundle adjustment step to achieve sub-pixel accurate camera models with comparable accuracy to COLMAP (under the conditions where COLMAP succeeds).  The metrics only measure if the camera rotation is within 15 degrees of correct angle and within 10% of the scene scale in position.

In terms of clarity of the work, one concern I have is that the authors often say "sparse-view" when it would be more accurate to say "wide-baseline".  For example, the abstract states that 3D reconstruction remains challenging for sparse views (<10).  It's not the reduced number of views that are the challenge, it's the wide baseline between those images.  More traditional methods like COLMAP would do just fine on 10 images from more similar viewpoints.

Also, I found the camera visualization in Figure 5-9 to be confusing.  Without the context of the 3D object or the coordinate system and with only a few cameras, it's hard to interpret what I'm looking at.  In many cases it's not even clear which cameras belong to which algorithm's results.

Minor issues:

Typo on middle of page 4: "the camera camera extrinsics"

References to Tab 10 and Tab 4 at the start of Section 4.3 seem to point to incorrect tables.

**Questions:**

I'm curious whether the authors think that this method would be effective in more realistic multi-view imaging environments where the imagery is not tightly cropped to just one object and where the camera motion could be more general?  Have any experiments been run to see if this method works on images "in the wild"?

In the the experiments, while is COLMAP used with SuperPoint features and SuperGlue matching?  No justification is given.  Is this expected to perform better or worse than vanilla COLMAP on this dataset?

---

> ### Author Response · Authors · 2023-11-16
>
> > This tight cropping ensures that most rays sampled pass through this common object in all views, which could provide added benefit to the propose approach. The cropping also provides a disadvantage to feature matching approaches such as used by COLMAP.
>
> We would like to clarify that for the COLMAP baseline, we use the entire image, and for the PoseDiffusion baseline, we center-crop the image (following their protocol). The reviewer is correct that our training with cropped images could provide additional contextual information, but we based our experimental setup on RelPose.
>
> > The metrics only measure if the camera rotation is within 15 degrees of correct angle and within 10% of the scene scale in position.
>
> In Tables 10 and 11, we analyze the rotation and camera center accuracies at a variety of thresholds. We have also added a new measure of rotation and camera center using AUC which evaluates accuracies at all thresholds in Tables 8 and 9, and visualize the threshold vs accuracy curve in Figure 11.
>
> > one concern I have is that the authors often say "sparse-view" when it would be more accurate to say "wide-baseline"
>
> We thank the reviewer for bringing up this discussion. We agree that “sparse-view” may not be the optimal term. However, even the term “wide-baseline” may not apply in all scenarios, e.g. 2 cameras that are spatially close but viewing different directions. Perhaps a more technically accurate term would be “sparsely sampled views,” e.g., if we think of a lightfield $L(x,y,u,v)$, sparse vs densely sampled views differ in the density of the sampling of $L$. We would be happy to emphasize this in the text, but if the reviewer does not have any objections, we would prefer to stick with the current title to follow convention in prior work (e.g. sparse-view reconstruction).
>
> > Also, I found the camera visualization in Figure 5-9 to be confusing. Without the context of the 3D object or the coordinate system and with only a few cameras, it's hard to interpret what I'm looking at. In many cases it's not even clear which cameras belong to which algorithm's results.
>
> We have re-made all the qualitative results figures starting from Figure 5. We respectfully ask the reviewer for feedback on the new presentation of results.
>
> > I'm curious whether the authors think that this method would be effective in more realistic multi-view imaging environments where the imagery is not tightly cropped to just one object and where the camera motion could be more general?
>
> Our method can generalize to such setups. Even though our method was trained with tight crops on object-centric CO3D, we find that it generalizes zero-shot to RealEstate10K even though we used center crops (following PoseDiffusion) and the dataset has different camera trajectories. Similarly, our MegaDepth experiments use center cropping and the dataset has dramatically different camera motion.
>
> > Have any experiments been run to see if this method works on images "in the wild"?
>
> We include results on self-captured data in Figure 6. We found that our method effectively generalizes to object-centric captures of object categories that are not found in CO3D.
>
> > In the the experiments, while is COLMAP used with SuperPoint features and SuperGlue matching? No justification is given. Is this expected to perform better or worse than vanilla COLMAP on this dataset?
>
> In our early experiments, we found that COLMAP with SuperPoint features and SuperGlue matching performs better than COLMAP with SIFT features and nearest neighbors matching on CO3D.

---

### Official Review · Reviewer_NPGi · 2023-11-01

**Soundness:** 3 good
**Presentation:** 2 fair
**Contribution:** 3 good
**Rating:** 8
**Confidence:** 4

**Summary:**

In this paper, authors tackle the problem of Sparse-View Pose Estimation by distributed "ray" based representation of cameras. By defining the camera as a bundle of rays via Plucker coordinates, authors formulate regression and diffusion-based approaches to predict camera rays from a set of sparse RGB images. With a predicted ray bundle, camera parameters (intrinsics and extrinsic) can be easily recovered. The authors test their method in a sparse view setup for the CO3Dv2 dataset and show that their regression and diffusion-based methods outperforms current learning-based and correspondence-based methods.

**Strengths:**

- I think this is a good method of formulating camera pose and intrinsic recovery using a bundle of rays. Furthermore, the authors' observation that ray-based representation is well-suited for set-level transformers is well backed by the results.
- The authors' "regression" based method outperforms other "diffusion" based methods, which shows that over-parameterization is really helping solve for camera geometry accurately.
- The results outperforms currently available "leaning" based and "correspondence" based method in sparse view settings on the CO3Dv2 dataset, that's a big encouragement.
- The authors also show that the method generalizes to out-of-distribution, in-the-wild scenes.

**Weaknesses:**

- One dataset is too small to see the applicability of a method. Since I see this method as superior to "PoseDiffusion", it would be great to see some results on the "scene-centric" dataset and compare it against PoseDiffusion.
- It would be nice to see a "memory" requirement to run these models. Processing N image features together, I am assuming requires a good amount of GPU memory.
- It would also be nice to see accuracy at different thresholds i.e. @5, @10, @15.
- It would also be nice to see an ablation study where we do not scale the poses. Most of the applications require properly "scaled" poses.

- The language is clear but I think the paper presentation is poor. Here are a few suggestions to improve the readability of the paper.
1) For e.g., Fig 2. is really confusing where the authors are trying to show the camera to ray-bundle and ray-bundle to camera process.
2) Fig 5. A qualitative comparison is hard to see and to make a good sense, as opposed to Fig 4 of PoseDiffusion paper for example.
3) Also good to say in eq (3) that "d" is obtained by unprojecting rays from camera pixel coordinates, and "m" is obtained by considering point "p" as the camera-center since all rays intersect at the camera center.
4) In section 4.3 evaluation Table numbers are wrong. Tab 10 -> Tab 1, Tab 4 -> Tab 2
5) Fig 6 is very confusing. I think this needs to be redone.

**Questions:**

- I see runtime in the appendix, but how does this method scale with adding a number of views? Instead of 8 what if I had 32? What are the memory requirements and inference runtime graphs?
- If I am interested in "scaled" poses, how would I get it?
- Authors say they stopped the backward diffusion process early and found that those estimates were better. How early did they stop the diffusion? And how did they choose when to stop? It would be nice to see some ablation analysis.
- What are your views of the applicability of this method for "scene" centric datasets?

---

> ### Author Response · Authors · 2023-11-16
>
> > It would be nice to see a "memory" requirement to run these models. Processing N image features together, I am assuming requires a good amount of GPU memory.
>
> We thank the reviewer for this suggestion. We have added a GPU memory analysis for our Ray Diffusion model to the paper in Table 7. According to `nvidia-smi`, backward diffusion using our ray diffusion model on 8 images reaches a peak GPU memory usage of 3095 MiB of VRAM, which should fit on any standard GPU.
>
> > It would also be nice to see accuracy at different thresholds i.e. @5, @10, @15.
>
> In Tables 10 and 11, we analyze the rotation and camera center accuracies at a variety of thresholds. We have also added a new measure of rotation and camera center using AUC which evaluates accuracies at all thresholds in Tables 8 and 9, and visualize the threshold vs accuracy curve in Figure 11.
>
> > It would also be nice to see an ablation study where we do not scale the poses. Most of the applications require properly "scaled" poses.
>
> We would like the reviewer to clarify what is meant by not scaling poses. We are interpreting the reviewer’s question as whether it’s possible to not normalize camera scale at training time and instead learn metric scale. In our camera normalization procedure, the sampled minibatches of cameras are re-scaled such that the first camera always has a unit translation. This procedure is done because the ground truth camera poses are acquired using COLMAP which can produce scenes in an arbitrary scene scale, which is not consistent between sequences. Thus, it is not possible to learn metric scale since the magnitude of translations would be ill-defined. If we have incorrectly interpreted the reviewer’s statement, we politely request clarification.
>
> > The language is clear but I think the paper presentation is poor. Here are a few suggestions to improve the readability of the paper.
>
> We thank the reviewer for the feedback and have made revisions accordingly.
> We have re-made all the qualitative results figures (Figure 5 onward). We respectfully ask the reviewer for feedback on the new presentation of results.
> For Figure 2 specifically, we are trying to convey 1) that the camera and ray representation are easily interchangeable and 2) that even noisy rays (e.g. if they do not intersect at a single point) can still be converted into a valid camera. We would appreciate any feedback from the reviewer on how to make this more clear. We have updated the caption to add more context.
>
> > I see runtime in the appendix, but how does this method scale with adding a number of views? Instead of 8 what if I had 32? What are the memory requirements and inference runtime graphs?
>
> Although our model is trained with 8 images, we find that it can generalize effectively to more images by re-sampling mini-batches during the backward diffusion process. Specifically, at each iteration of DDPM, we can sample batches of 8 images (keeping the first image fixed), and predict the $X_0$s for each batch separately. With more images, the memory cost is constant but the runtime scales linearly. We find that our performance remains consistent with more images despite training with only 8 images, as we report below and added to Table 5 in the paper
>
> | # of Images                           | 8    | 15   | 22   | 29   | 36   | 43   |
> |----------------------------------------|------|------|------|------|------|------|
> | Rotation Acc. (Seen Categories)        | 93.3 | 93.1 | 93.3 | 93.1 | 93.4 | 93.4 |
> | Rotation Acc. (Unseen Categories)      | 88.1 | 88.2 | 89.2 | 88.7 | 89.0 | 88.9 |
> | Camera Center Acc. (Seen Categories)   | 84.1 | 78.3 | 76.5 | 75.3 | 74.7 | 74.2 |
> | Camera Center Acc. (Unseen Categories) | 71.4 | 62.7 | 61.1 | 59.3 | 59.2 | 58.9 |
>
>
>
> > What are your views of the applicability of this method for "scene" centric datasets?
>
> Please refer to the top-level response. We have added experiments on zero-shot generalization to RealEstate10K (following PoseDiffusion) as well as newly trained results on MegaDepth.
>
> > Authors say they stopped the backward diffusion process early and found that those estimates were better. How early did they stop the diffusion? And how did they choose when to stop? It would be nice to see some ablation analysis.
>
> In our initial experiments, we found that our accuracy metrics generally peak around T=30 (where T=100 is complete noise and T=0 is the “normal” final diffusion timestep). We have added an ablation that shows performance at all timesteps in Figure 12.
>
> > I found the camera visualization in Figure 5-9 to be confusing.
>
> We have re-made all the qualitative results figures (Figure 5 onward). We respectfully ask the reviewer for feedback on the new presentation of results.

---

> > ### Comment · Reviewer_NPGi · 2023-11-22
> > **Thanks for the clarifications**
> >
> > I thank authors for putting efforts in a nice rebuttal.
> >
> > Regarding scaled poses, I guess I misunderstood what you meant. I thought all translations were scaled to be unit norm vectors, and we were only getting relative distance. If we scale only the first pose to be unit-norm translation vector, its pretty standard.
> >
> > Since really basic oracle works better than any of the method on RealEstate10k, I can't really use it to rate the generalization of this method.
> > Thanks for providing results on MegaDepth though, any reason you don't have a numerical comparison? I see that SP+SG is more accurate when converges. When it doesn't converge, there are many heuristics one can apply, such as image retrieval and then coarse matching, is Ray Diffusion more accurate than such heuristics?
> >
> > Thanks for ablation on memory requirements, runtimes, accuracy at different threshold.
> >
> > Overall, I am inclined to put this in acceptance threshold. My only concern is that there should have been numerical comparison for this method on scene centric dataset (not RealEstate10k because of bias), and it should really be part of the main paper (not the appendix). The idea of ray diffusion definitely shows novelty in a limited setting, hence I would be keeping my rating at acceptance. Had there been results on multiple datasets, my ratings would be in strong acceptance.

---

> > > ### Author Response · Authors · 2023-11-23
> > >
> > > We would like to thank the reviewer for their response.
> > >
> > > First, we would like to make a minor clarification – our approach does outperform the naive baseline on RealEstate10k, reducing the fractions of errors by a third (error rate from ~16% to ~11%).
> > >
> > > Regarding MegaDepth, we note that this dataset comprises of large-scale scenes, and sampling a few random images per scene can lead to uncorrelated viewpoints (e.g. images from different city blocks with no visual overlap!). We believe that this setup (i.e. naive random sampling of a few images for a large scene) is not a practically relevant one, and one may need more careful consideration to define a useful empirical protocol for evaluation (e.g. consider sets of views with overlapping visual frustums, or limit view sampling within a spatial grid, etc.). For the purpose of demonstration, we simply trained by naive sampling and demonstrated results on selected sequences where the images did have visual overlap, but this is not an experimental protocol we wish to recommend to the community (and possibly set a bad precedent). In fact, the issues with RealEstate10K (a benchmark proposed in previous published work) highlight the challenge of developing a healthy scene-centric benchmark.
> > >
> > > Finally, regarding possible heuristics (e.g. image retrieval and coarse matching), these are more relevant when given a very large number of images e.g. given 100s of images of the Colosseum, one can retrieve similar ones. In our setup with only a few images (<=8), these are not easily applicable.

---

### Author Response · Authors · 2023-11-16

We thank all reviewers for their thoughtful comments and valuable feedback. Here, we address comments that are relevant to all reviewers. Reviewer-specific comments are left as direct comments.

All reviewers note that the evaluation setup is limited to training and testing on turntable-style sequences from the CO3D dataset. Reviewers also wondered if our approach is applicable to scene-level datasets. This point is well taken. Following PoseDiffusion, we first test the *zero-shot* generalization to RealEstate10K, a scene-level dataset. We report these results in Table 4, and also include them here:

| Rotation Acc         | 2             | 3             | 4             | 5             | 6             | 7             | 8             |
|-----------------------|---------------|---------------|---------------|---------------|---------------|---------------|---------------|
| Constant Rot.         | 84.0          | 83.8          | 83.9          | 84.0          | 84.0          | 84.0          | 83.9          |
| PoseDiffusion         | 77.6          | 77.9          | 78.4          | 78.7          | 78.9          | 79.3          | 79.0          |
| RelPose++             | 83.8          | 85.1          | 85.8          | 86.4          | 86.5          | 86.7          | 86.8          |
| Ray Regression (Ours) | 90.8          | **90.0** | **89.9** | **89.7** | **89.5** | **89.5** | **89.5** |
| Ray Diffusion (Ours)  | **90.9** | 89.9          | 89.5          | 89.3          | 89.1          | 88.8          | 88.3          |

| Cam Center Acc           | 2             | 3             | 4             | 5             | 6             | 7             | 8             |
|-----------------------|---------------|---------------|---------------|---------------|---------------|---------------|---------------|
| PoseDiffusion         | 100           | 77.7          | 65.9          | 60.1          | 55.0          | 52.2          | 50.2          |
| RelPose++             | 100           | 71.2          | 60.6          | 54.0          | 49.4          | 47.1          | 45.5          |
| Ray Regression (Ours) | 100           | 74.4          | 62.0          | 56.0          | 51.3          | 49.2          | 47.1          |
| Ray Diffusion (Ours)  | 100           | **79.7** | **68.6** | **62.2** | **57.8** | **54.9** | **52.1** |


All methods are trained only on CO3D. We found that our method outperforms existing approaches on this task setup. However, we note that even a naive baseline that only predicts a constant identity rotation (Constant Rot.) performs rather well, suggesting that the dataset has a strong forward-facing bias.

Following Reviewer mia4’s suggestion, we re-train our model on the MegaDepth SfM dataset. We find that our method can successfully recover meaningful camera poses from 8 wide-baseline views on held-out sequences (see Figure 10). We believe that this result is perhaps more illuminating of the generalizability of our approach on scene-level datasets.

Finally, please note that we have also updated the results for our method in Tables 1 and 2 since submission time. We fixed a minor bug in training where we were incorrectly adjusting the camera intrinsics based on the image crop. Fixing this led to a slight improvement in our method across the board.

Once again, we would like to thank the reviewers for their helpful comments and look forward to improving the paper with the suggestions.

---

### Meta-Review · Area_Chair_AMTC · 2023-12-07

**Metareview:**

The paper addresses the problem of estimating camera pose from a small collection of wide-baseline images.  The paper makes very novel and inventive use of an old idea (cameras as bundles of rays) and shows that this representation, previously used in a quite different context, is a good fit for modern machine learning models.

Strengths:
 - All reviewers agree that the use of a ray-bundle representation is novel in this context.  To this AC, it is one of the more novel ideas in some time in this area.

Weaknesses:
 - All reviewers agree that it would be desirable to include much more experimental exploration: on more datasets, against more baselines.

The reviewers make many excellent suggestions which the authors have already incorporated.  I would echo the thoughts about "wide baseline" versus "sparse view".  I believe that "sparse view" was coined in [A,2022], as the citations therein refer to papers which use "wide baseline" as the term.  So yes, it is a new and better name, but it might need some text relating it to "wide baseline" which is more established.

[A] PlaneFormers: From Sparse View Planes to 3D Reconstruction, Samir Agarwala, Linyi Jin, Chris Rockwell, and David F. Fouhey, ECCV 2022

[Aside, not relevant to decision process: Given that the ray-bundle idea was initially intended for use with complex camera geometries (e.g. fisheye or more exotic lenses), it might be interesting to see how the technique works with such geometries, e.g. a fisheye camera passing through an environment.]

**Justification For Why Not Higher Score:**

n/a

**Justification For Why Not Lower Score:**

This is one of those papers where the idea is completely obvious *after* one has been told it, and thus it deserves widespread dissemination.  The only reason not to vote it higher is that, as the reviewers suggest, it has been shown to work only on one dataset.
Therefore presentation as an oral paper might prompt an audience to feel the paper is being unfairly rewarded for the idea, when more work needs to be done to prove that the idea is a good one.

---

### Decision · Program_Chairs · 2024-01-16

Accept (oral)